# RETHINKING TEST-TIME LIKELIHOOD: THE LIKELIHOOD PATH PRINCIPLE AND ITS APPLICATION TO OOD DETECTION

## ABSTRACT

While likelihood is attractive in theory, its estimates by deep generative models (DGMs) are often broken in practice, and perform poorly for out of distribution (OOD) Detection. Various recent works started to consider alternative scores and achieved better performances. However, such recipes do not come with provable guarantees, nor is it clear that their choices extract sufficient information.

We attempt to change this by conducting a case study on variational autoencoders (VAEs). First, we introduce the *likelihood path (LPath) principle*, generalizing the likelihood principle. This narrows the search for informative summary statistics down to the *minimal sufficient statistics* of VAEs' conditional likelihoods. Second, introducing new theoretic tools such as *nearly essential support*, *essential distance* and *co-Lipschitzness*, we obtain non-asymptotic provable OOD detection guarantees for certain distillation of the minimal sufficient statistics. The corresponding LPath algorithm demonstrates SOTA performances, even using simple and small VAEs [1] with poor likelihood estimates. To our best knowledge, this is the first provable unsupervised OOD method that delivers excellent empirical results, better than any other VAEs based techniques.

## 1 INTRODUCTION

Independent and identically distributed (IID) samples in training and test times is the key to much of machine learning (ML)'s success. For example, this experimentally validated modern neural nets before tight learning theoretic bounds are established. However, as ML systems are deployed in the real world, out of distribution (OOD) data are apriori unknown and pose serious threats. This is particularly so in the most general setting where labels are absent, and test input arrives in a streaming fashion. While attractive in theory, naive approaches, such as using the likelihood of deep generative models (DGMs), are proved to be ineffective, often assigning high likelihood to OOD data (Nalisnick et al., 2018). Even with access to perfect density, likelihood alone is still insufficient to detect OOD data Le Lan & Dinh (2021); Zhang et al. (2021) when the IID and OOD distributions overlap.

In response to likelihood's weakness, most works have focused on either improving density models Havtorn et al. (2021); Kirichenko et al. (2020) or taking some form of likelihood ratios with a baseline model chosen with prior knowledge about image data (Ren et al., 2019; Serrà et al., 2019; Xiao et al., 2020). Recent theoretical works (Behrmann et al., 2021; Dai et al.) show that perfect density estimation may be infeasible for many DGMs. It is thus logical to consider OOD screening scores that are more robust to density estimation, following Vapnik's principle de Mello & Ponti (2018): *When solving a problem of interest (OOD detection), do not solve a more general problem (perfect density estimation) as an intermediate step.* Some recent works on OOD detection Ahmadian & Lindsten (2021); Bergamin et al. (2022); Morningstar et al. (2021); Graham et al. (2023); Liu et al. (2023) indeed start to consider other information contained in the entire neural activation path leading to the likelihood. Examples include entropy, KL divergence, and Jacobian in the likelihood Morningstar et al. (2021). See Section A.1 for more discussions on related works. However, it is not obvious what kind of statistical inferences these statistics perform, nor do they come with provable

---

[1]We use the same model as Xiao et al. (2020), open sourced from: `https://github.com/XavierXiao/Likelihood-Regret`.

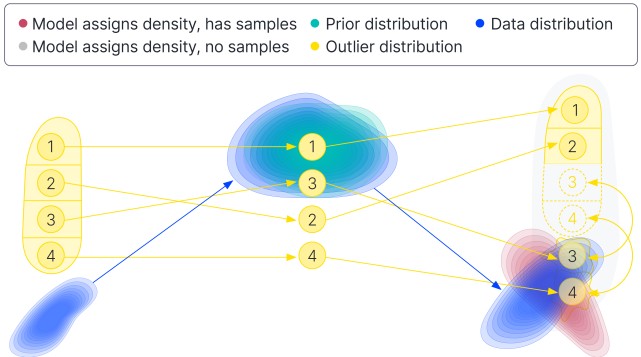

Figure 1: **Main idea illustration. Left**, we have $\mathbf{x}_{\text{IID}}$ distribution (blue) and $\mathbf{x}_{\text{OOD}}$ distribution (yellow) in the visible space. $\mathbf{x}_{\text{OOD}}$ is classified into four cases. **Middle**, we have prior (turquoise), posterior after observing $\mathbf{x}_{\text{IID}}$ (blue), posterior divided into four cases after observing $\mathbf{x}_{\text{OOD}}$ (yellow), in the latent space. **Right**, we have the reconstructed $\widehat{\mathbf{x}}_{\text{IID}}$ (red) on top of real $\mathbf{x}_{\text{IID}}$ distribution (blue), and $\widehat{\mathbf{x}}_{\text{OOD}}$ again divided into four cases. Cases (1) and (2)'s graphs means $\widehat{\mathbf{x}}_{\text{OOD}}$ is well reconstructed, while the fried egg alike shapes for Cases (3) and (4) indicate $\widehat{\mathbf{x}}_{\text{OOD}}$ are poorly reconstructed. The grey area indicates some pathological OOD regions where VAE assigns high density but not a lot of volume. When integrated, these regions give nearly zero probabilities, and hence the data therein cannot be sampled in polynomial times. These are *atypical sets*.

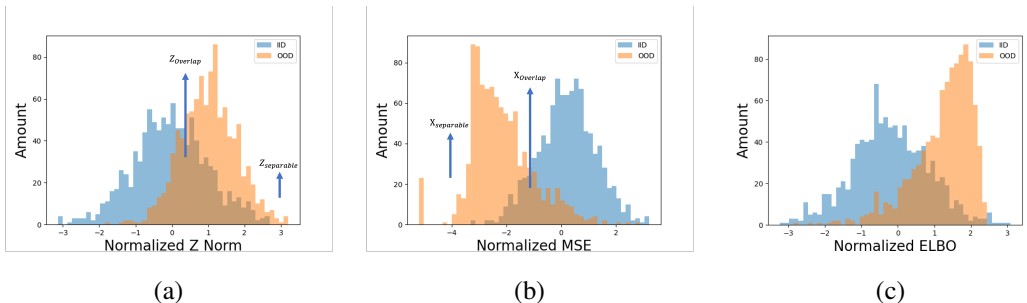

| (a) | (b) | (c) |

Figure 2: Example histograms of (a): Z Norm $\|\mathbf{z}\|_2$, corresponding to the prior $p\left(\mathbf{z}^k\right)$ in equation 1. (b): MSE, $\|\mathbf{x} - \widehat{\mathbf{x}}\|_2$, corresponding to the conditional likelihood $p_\theta\left(\mathbf{x} \mid \mathbf{z}^k\right)$ in equation 1. (c): ELBO, corresponding to $\log p_\theta(\mathbf{x})$ in equation 1. The four cases in Figure 1 can be mapped to combinations of $Z_{\text{overlap}}$, $Z_{\text{separable}}$ in (a) and $X_{\text{overlap}}$, $X_{\text{separable}}$ in (b). Case (1): $Z_{\text{overlap}} + X_{\text{overlap}}$; Case (2): $Z_{\text{separable}} + X_{\text{overlap}}$; Case (3): $Z_{\text{overlap}} + X_{\text{separable}}$; Case (4): $Z_{\text{separable}} + X_{\text{separable}}$. The separation is less pronounced in ELBO. IID data is CIFAR10, OOD data is SVHN. $\mathbf{z}$ is 100 dimensional. All values are normalized, for details on normalization, see Appendix C.

guarantees. To sum, while the entire neural activation path contains all the information, it is hard to choose which statistics for test time inferences, with theoretical guarantees.

Understanding OOD detection's theoretical limitation is arguably more important than the IID settings, because OOD data are unknown in advance which makes the experimental validation less reliable than the IID cases. However, very few works (except Fang et al. (2022)) explore the theory of the OOD detection, especially in the general unsupervised case. This paper makes a theoretical step towards changing this. We develop a *principled* and *provable* method, and show state-of-the-art (SOTA) OOD detection performance can be achieved using simple and small VAEs with poor likelihood estimates. To clearly demonstrate the multi-fold contribution of this paper, we discuss the contributions from three perspectives: *empirical*, *methodological*, and *theoretical* ones.

**Empirical contribution.** We contribute a recipe (Section 4) for selecting OOD screening statistics, exploiting VAEs' structure (Figure 1). The recipe starts from this counter-intuitive question: for OOD detection, since practical VAEs are broken (Behrmann et al., 2021; Dai et al.), can we identify VAEs that are sub-optimal in the right way (instead of aiming for perfect density estimation) to achieve good performance? We give one positive answer. Our algorithm broadly follows DoSE Morningstar et al.

(2021)'s framework, but differs in two important aspects. First, our statistics perform explicit *instance dependent* inferences, allowing neural latent models (e.g. VAEs) to access rich literature in parametric statistical inferences (Section 2, Appendix B.5). Second, our choice, the *minimal sufficient statistics* of the encoder and decoder's conditional likelihoods, can provably detect OOD samples, even under imperfect estimation (Theorem 3.8). Our simple method delivers SOTA peformances (Table 1). We achieve so with DC-VAEs from Xiao et al. (2020)'s repository, which is much less powerful (in terms of parameter count) and much less well estimated (with regards to its generative sample quality). We believe this "achieving more with less" phenomenon proves our method's potential.

**Methodological contribution.** The aforementioned recipe follows our newly proposed *likelihood path principle* (LPath) which generalizes the classical likelihood principle [2]: when performing instance dependent inference (e.g. OOD detection) under imperfect density estimation, more information can be obtained from the neural activation path that estimates $p_\theta(\mathbf{x})$. Note that the search space is much smaller, by not considering arbitrary functions of activation. We only consider the activation that propagate to $p_\theta(\mathbf{x})$. We believe this principle is of independent interests to representation learning. If it is possible to extend it to more powerful models (e.g. Glow Kingma & Dhariwal (2018) or diffusion models Rombach et al. (2022)), we anticipate better results. This is left to future works.

**Theoretical contribution.** In the general unsupervised OOD detection literature, ours is the first work that quantifies how well VAEs can screen OOD (Theorem 3.8) to our best knowledge. To prove such results, we introduce *nearly essential support*, *essential separation* and *essential distance* (Definitions 3.1, 3.2, 3.3, 3.4) for distributions, capturing both near-OOD and far-OOD cases (Fang et al., 2022). We also generalize Lipschtiz continuity and injectivity (Definitions 3.6, B.6, B.7) to describe how VAEs detect OOD samples. These new concepts that describe the encoder and decoder's function analytic properties, the essential distance between $P_{\text{IID}}$ and $P_{\text{OOD}}$, as well as VAEs' test time reconstruction error characterize our method. Our argument is combinatorial and geometric in nature, which complements the traditional statistical and information theoretic tools.

The rest of the paper is organized as follows. Section 2 bases our method on well established statistical principles. Section 3 details our theory. Section 4 describes our algorithm and 5 presents an empirical evaluation of our algorithm and shows that our proposed LPath method achieves SOTA in the widely accepted unsupervised OOD detection benchmarks.

## 2 FROM THE LIKELIHOOD PRINCIPLE TO THE LIKELIHOOD PATH PRINCIPLE

This section discusses the statistical foundation of our *likelihood path principle*. We begin with suboptimality in existing methods (*problem I* and *problem II*), followed by proposing the minimal sufficient statistics of VAEs' conditional likelihoods as a solution.

**Problem I: VAEs' encoder and decoder contain complementary information for OOD detection, but they can be cancelled out in** $\log p_\theta(\mathbf{x})$. Recall VAEs' likelihood estimation:

$$\log p_\theta(\mathbf{x}) \approx \log \left[ \frac{1}{K} \sum_{k=1}^{K} \frac{p_\theta\left(\mathbf{x} \mid \mathbf{z}^k\right) p\left(\mathbf{z}^k\right)}{q_\phi\left(\mathbf{z}^k \mid \mathbf{x}\right)} \right], \tag{1}$$

which aggregates both lower and higher level information. The decoder $p_\theta\left(\mathbf{x} \mid \mathbf{z}^k\right)$'s reconstruction focuses on the pixel textures, while encoder $q_\phi\left(\mathbf{z}^k \mid \mathbf{x}\right)$'s samples evaluated at the prior, $p\left(\mathbf{z}^k\right)$, describe semantics. Consider $\mathbf{x}_{\text{OOD}}$, whose lower level features are similar to IID data, but is semantically different. We can imagine $p_\theta\left(\mathbf{x} \mid \mathbf{z}^k\right)$ is large while $p\left(\mathbf{z}^k\right)$ is small. However, (Havtorn et al., 2021) demonstrates $p_\theta(\mathbf{x})$ is dominated by lower level information. Even if $p\left(\mathbf{z}^k\right)$ wants to reveal $\mathbf{x}_{\text{OOD}}$'s OOD nature, we cannot decipher it through $p_\theta(\mathbf{x}_{\text{OOD}})$. The converse: $p_\theta\left(\mathbf{x} \mid \mathbf{z}^k\right)$ can flag $\mathbf{x}_{\text{OOD}}$ when the reconstruction error is big. But if $p\left(\mathbf{z}^k\right)$ is unusually high compared to typical $\mathbf{x}_{\text{IID}}$, $p_\theta(\mathbf{x})$ may appear less OOD. We illustrate the main idea with Fig. 1 and demonstrate the four cases with histograms from real data in Fig. 2. See Section 3.2 for an in-depth analysis and Table 1 for some empirical evidence. To conclude, useful information for screening $\mathbf{x}_{\text{OOD}}$ is diluted in either case, due to the *arithmetical cancellation* in multiplication (experimentally verified in Table 3).

**Problem II: Too much overwhelms, too little is insufficient**. On the other spectrum, one may propose to track *all* neural activations. Since this is not tractable, Morningstar et al. (2021) carefully

---

[2]The marginal likelihood $p_\theta(\mathbf{x})$ is a special case, because it only uses the end point in the likelihood path.

selects various summary statistics. But it is unclear whether they contain sufficient information. Moreover, these approaches require fitting a second stage classical statistical algorithm on the chosen statistics, which typically work less well in higher dimensions (Maciejewski et al., 2022). Without a principled selection, including too many can cripple the second stage algorithm; having too few loses critical information. Neither extreme (tracking too many or too few) seems ideal.

**Proposed Solution: The Likelihood Path Principle.** We propose and apply our *likelihood path principle* to VAEs. This entails applying the *likelihood principle* twice in VAEs' encoder and decoder distributions, and track their *minimal sufficient statistics*: $T(\mathbf{x}, \mathbf{z}^k) = (\mu_{\mathbf{x}}(\mathbf{z}^k), \sigma_{\mathbf{x}}(\mathbf{z}^k), \mu_{\mathbf{z}}(\mathbf{x}), \sigma_{\mathbf{z}}(\mathbf{x}))$. We then fit a second stage statistical algorithm on them, akin to Morningstar et al. (2021). We refer to such sufficient statistics as VAEs' *likelihood paths* and name our method the *LPath* method. Our work differs from others in two major ways. First, our choices are based on the well established likelihood and sufficiency principles, instead of less clear criteria. Second, our method can remain robust to imperfect $p_\theta(\mathbf{x})$ estimation, provably (Theorem 3.8).

**Instance-dependent parametric inference opens door for neural nets to rich methods from classical statistics**. When $p_\theta(\mathbf{x} \mid \mu_{\mathbf{x}}(\mathbf{z}^k), \sigma_{\mathbf{x}}(\mathbf{z}^k))$ and $q_\phi\left(\mathbf{z}^k \mid \mu_{\mathbf{z}}(\mathbf{x}), \sigma_{\mathbf{z}}(\mathbf{x})\right)$ are Gaussian parameterized, the inferred *instance dependent* parameters $T(\mathbf{x}, \mathbf{z}^k)$ allow us to perform statistical tests in both latent and visible spaces. By the no-free-lunch principle in statistics[3], this model-specific information can be advantageous versus generic tests [4] based on $p_\theta(\mathbf{x})$ alone. By the *likelihood principle*, which states that in the inference about model parameters, after data is observed, all relevant information is contained in the likelihood function. Thus $T(\mathbf{x}, \mathbf{z}^k)$ is sufficiently informative for OOD inferences. Unlike classical statistical counterparts, which are often static, $T(\mathbf{x}, \mathbf{z}^k)$ is dynamic depending on neural activation. However, they still inherit the inferential properties, capturing all information in the sense of the well established *likelihood and sufficiency principles*. In the VAEs' case, LPath is built by $q_\phi(\mathbf{z} \mid \mathbf{x})$, $p(\mathbf{z})$, and $p_\theta(\mathbf{x} \mid \mathbf{z})$, which depends on $T(\mathbf{x}, \mathbf{z}^k)$. This LPath can surprisingly benefit when *VAEs break in the right way* (Appendix B.4.3). Our *likelihood path principle* generalizes the likelihood principle, by considering the neural activation path that leads to $p_\theta(\mathbf{x})$. Greater details are discussed in Section B.5.

Modern DGMs are very powerful, but their complexity prevents them from having closed form sufficient statistics in the $p_\theta(x)$. As such, it is unclear how to apply the likelihood and sufficiency principles. While VAEs don't even compute $p_\theta(x)$ exactly, its encoder-decoder LPath infers instance-dependent parameters which are minimal sufficient statistics. For this reason, it is an ideal candidate to test the likelihood path principle. Our analysis centers around it in this paper.

## 3 FROM THE LIKELIHOOD PATH PRINCIPLE TO OOD DETECTION

In Section 2, we narrowed the search of a good OOD detection recipe, from all possible activation paths down to VAEs' minimal sufficient statistics: $T(\mathbf{x}, \mathbf{z}^k) = \mu_{\mathbf{x}}(\mathbf{z}^k), \sigma_{\mathbf{x}}(\mathbf{z}^k), \mu_{\mathbf{z}}(\mathbf{x}), \sigma_{\mathbf{z}}(\mathbf{x})$. However, two issues remain. First, they remain high dimensional. This not only costs computational time, but can also cause trouble to the second stage statistical algorithm (Maciejewski et al., 2022). Second, while they are based on statistical theories, they don't come with OOD detection performance guarantee, ideally depending on *datasets, VAEs' functional and statistical generalization properties*. This section complements our statistical principles with rigorous *non-asymptotic bounds*. Generalizing point-wise injectivity and Lipschitz continuity, Section 3.1 develops new tools to establish data and model dependent bounds on VAEs OOD detection (Theorem 3.8). Aided by these inequalities, Section 3.2 finalizes the OOD detection algorithm by combining statistical and geometric theories.

### 3.1 PROVABLE DATA AND MODEL DEPENDENT OOD DETECTION PERFORMANCES

In Section 3.1.1, we introduce essential separation and distances (Definition 3.1, 3.2 3.3). Section 3.1.2 generalizes injectivity and Lipschitz continuity. These are relevant for OOD detection as they can describe how VAEs can mix $P_{\text{IID}}$ and $P_{\text{OOD}}$ together in both the visible and latent spaces. These new tools are not VAEs specific and can be of independent interests for general representation learning. Using such concepts, Section 3.1.3 proves how well VAEs' minimal sufficient statistics

---

[3]Tests which strive to have high power against all alternatives (model agnostic) can have low power in many important situations (model specific), see Simon & Tibshirani (2014) for another concrete example.

[4]For example, typicality test in Nalisnick et al. (2019) and likelihood regret in Xiao et al. (2020)

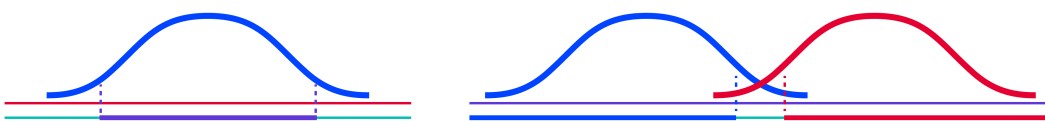

Figure 3: **Left:** $\text{Supt}(P_{\text{IID}})$ is the red solid line, which is decomposed to one **nearly essential support** (purple solid line), and less likely events (two green solid lines). **Right:** $\text{Supt}(P_{\text{IID}})$ and $\text{Supt}(P_{\text{OOD}})$ are the purple solid line. $\text{ESupt}(P_{\text{IID}})$ is the blue solid line and $\text{ESupt}(P_{\text{OOD}})$ is the red solid line. The green solid line depicts the corresponding **essential distance**, so they are **essentially separable**. The key idea is that for many overlapped distributions, most of their samples are separable.

can detect OOD depending on: 1. how separable $P_{\text{IID}}$ and $P_{\text{OOD}}$ are in the visible space; 2. how well the decoder reconstructs $P_{\text{IID}}$; 3. how badly encoder $q_\phi$ confuse between $P_{\text{IID}}$ and $P_{\text{OOD}}$ in the latent space; 4. how Lipscchitz continuous the decoder $p_\theta$ is.

### 3.1.1 Essential Separation and Essential Distance

We introduce a class of *essential separation* concepts below. They are applicable to both the far-OOD and near-OOD cases (Fang et al., 2022; Hendrycks & Gimpel, 2016; Fort et al., 2021). The high level idea is that, many $P_{\text{IID}}$ and $P_{\text{OOD}}$ pairs are separable if we consider the more likely samples.

**Definition 3.1** (Nearly essential support of a Distribution). Let $P$ be a probability distribution with support $\text{Supt}(P)$ (See Appendix B.1 for a definition.) and $0 \le \epsilon < 1$ be given. We say a subset $\text{ESupt}(P; -\epsilon) \subset \text{Supt}(P)$ is *an $\epsilon$ nearly essential support[5] of $P$*, if $P(\text{ESupt}(P; -\epsilon)) \ge 1 - \epsilon$.

We omit $\epsilon$ when the context is clear. Intuitively, when $\epsilon$ is small, the subset $\text{ESupt}(P; -\epsilon)$ contains most events except those occurring with probability less than $\epsilon$. A pictorial illustration is shown on the left in Figure 3 and examples are in Section B.1. Among such nearly essential supports between $P_{\text{IID}}$ and $P_{\text{OOD}}$, we are interested in the ones that are maximally separable.

**Definition 3.2** (Essential Distance). Let $P_{\text{IID}}$ and $P_{\text{OOD}}$ be two probability distributions with supports in a metric space $(X, d_X)$, $\epsilon_{\text{IID}} \ge 0$ and $\epsilon_{\text{OOD}} \ge 0$ be given. We define the $(\epsilon_{IID}, \epsilon_{OOD})$ *essential distance* between the two distributions as:

$$D_{X|\epsilon_{\text{IID}}, \epsilon_{\text{OOD}}}(P_{\text{IID}}, P_{\text{OOD}}) \tag{2}$$

$$:= \sup_{\substack{\text{ESupt}(P_{\text{IID}}; -\epsilon_{\text{IID}}) \subset P_{\text{IID}} \\ \text{ESupt}(P_{\text{OOD}}; -\epsilon_{\text{OOD}}) \subset P_{\text{OOD}}}} d_X(\text{ESupt}(P_{\text{IID}}; -\epsilon_{\text{IID}}), \text{ESupt}(P_{\text{OOD}}; -\epsilon_{\text{OOD}})) \tag{3}$$

We believe this captures many practical cases much better. See also the right of Figure 3 for a graphical demonstration and Appendix B.1 for more examples. We can now define essential separability:

**Definition 3.3** (Essentially Separable between IID and OOD). Let $P_{\text{IID}}$ and $P_{\text{OOD}}$ be two probability distributions and $m_{\text{inter}} > 0$ [6] be given. We say $P_{IID}$ and $P_{OOD}$ are $(\epsilon_{IID}, \epsilon_{OOD})$ *essentially separable by $m_{inter}$*, if there exist $\epsilon_{\text{IID}} > 0$ and $\epsilon_{\text{OOD}} > 0$ such that:

$$D_{X|\epsilon_{\text{IID}}, \epsilon_{\text{OOD}}}(P_{\text{IID}}, P_{\text{OOD}}) \ge m_{\text{inter}} \tag{4}$$

$D_{X|\epsilon_{\text{IID}}, \epsilon_{\text{OOD}}}(P_{\text{IID}}, P_{\text{OOD}})$ depends on where and how much we remove certain events. Therefore, it can still provide a meaningful separation even when $\text{Supt}(P_{\text{IID}}) = \text{Supt}(P_{\text{OOD}})$. In turn, $(\epsilon_{\text{IID}}, \epsilon_{\text{OOD}})$ depends on the intrinsic level of separation between $P_{\text{IID}}$ and $P_{\text{OOD}}$. See Appendix B.1 for a measure theoretic view on our construction. We next relate $(\epsilon_{\text{IID}}, \epsilon_{\text{OOD}})$ to the essential distance/margin:

**Definition 3.4** (Margin Essential Distance). Under the setting in Definition 3.3, we define the margin $m_{\text{inter}}$ minimal support probabilities as the $\arg\min$ [7] for the following minimization problem:

$$\epsilon^*_{\text{IID}}, \epsilon^*_{\text{OOD}} = \arg \inf_{\substack{\epsilon_{\text{IID}}, \epsilon_{\text{OOD}} \ge 0 \quad \text{such that} \\ D_{X|\epsilon_{\text{IID}}, \epsilon_{\text{OOD}}}(P_{\text{IID}}, P_{\text{OOD}}) \ge m_{\text{inter}}}} \epsilon_{\text{IID}} + \epsilon_{\text{OOD}} \tag{5}$$

---

[5] We add the term nearly to avoid collision with the closely related *essential support* in real analysis.

[6] This margin is interpreted as the desired level of essential inter-distribution separation.

[7] Without loss of generality, if the $\arg\min$ does not exist, we consider $\epsilon^*_{\text{IID}}, \epsilon^*_{\text{OOD}}$ up to a desired level of precision. Among them, we choose one as an approximate minimum. The construction remains well-posed.

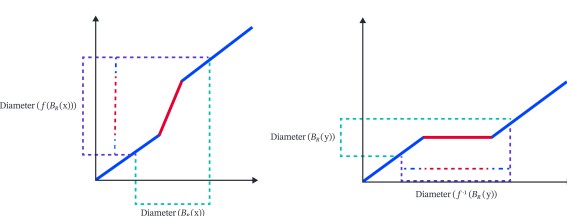

Figure 4: **Left**: If $f$ is $L$-Lipschitz, it cannot (forward) push one small region $B_R(\mathbf{x})$ to a big one (diameter no more than $2LR$) - $f$ is not "one-to-many". **Right**: If $f$ is $(K, k)$ co-Lipschitz, its preimage $f^{-1}$ cannot (backward) pull one small region $B_R(\mathbf{y})$ to a big one (diameter no more than $2KR + k$) - $f$ is "one-to-one".

The corresponding distance is called $m_{inter}$ *margin mini-max essential distance*:

$$\mathrm{D}_{X|m_{\mathrm{inter}}}(P_{\mathrm{IID}}, P_{\mathrm{OOD}}) = \mathrm{D}_{X|\epsilon_{\mathrm{IID}}^*, \epsilon_{\mathrm{OOD}}^*}(P_{\mathrm{IID}}, P_{\mathrm{OOD}}) \tag{6}$$

By construction, $\mathrm{D}_{X|m_{\mathrm{inter}}}(P_{\mathrm{IID}}, P_{\mathrm{OOD}}) \geq m_{\mathrm{inter}}$. Because of the union bound, we also say *with probability at least* $1 - (\epsilon_{IID}^* + \epsilon_{OOD}^*)$, $P_{IID}$ *and* $P_{OOD}$ *are separated by margin* $m_{inter}$.

### 3.1.2 GENERALIZING LIPSCHITZNESS

Next, we review classical point-wise injectivity and Lipschitz continuity, and then extend them into new ones. These geometric function analytic properties describe how encoder $q_\phi$ can confuse $P_{\mathrm{OOD}}$ to be $P_{\mathrm{IID}}$ in the latent space, and how well decoder $p_\theta$ can reconstruct $P_{\mathrm{OOD}}$ undesirably.

**Definition 3.5** (L-Lipschitz: region-wise not "one-to-many"). Let $(X, \mathrm{d}_X, \mu)$ and $(Y, \mathrm{d}_Y, \nu)$ be two metric-measure spaces, with equal (probability) measures $\mu(X) = \nu(Y)$. Let $L > 0$ be fixed. A function $f : X \to Y$ is $L$-**Lipschitz**, if for any $R \geq 0$, any $x \in X$:

$$\mathrm{Diameter}(f(B_R(x))) \leq L \cdot \mathrm{Diameter}(B_R(x)) \tag{7}$$

The equivalence between the geometric version and the standard L-Lipscthiz definition, along with more discussions, are in Appendix B.2. It is relevant for OOD detection, since how well decoder $p_\theta$ can reconstruct $P_{\mathrm{OOD}}$ depends on $p_\theta$'s Lipschiz constant, demonstrated by Case 1 in Figure 1 and Section 3.2. Point-wise **injectivity** (one-to-one), which dictates that $f^{-1}(y)$ *is singleton*, is a counterpart to continuity in the sense of invariance of dimension Müger (2015). However, this definition does not measure how "one-to-one", nor does it apply to a region. We quantitatively extend it to regions with positive probabilities, which may better suit probabilistic applications.

**Definition 3.6** (Co-Lipschitz: region-wise "one-to-one"). Let $K > 0, k \geq 0$ be given. Under the same settings as Definition 3.5, a function $f : X \to Y$ is **co-Lipschitz** with degrees $(K, k)$, if for any $y \in Y$, any $R \geq 0$:

$$\mathrm{Diameter}(f^{-1}(B_R(y))) \leq K \cdot \mathrm{Diameter}(B_R(y)) + k \tag{8}$$

We call it co-Lipschitz, because it is reminiscent to Definition 3.5, with $f(B_R(y))$ (forward mapping) replaced by $f^{-1}(B_R(y))$ (backward inverse image). Its relation to OOD detection is illustrated in Case 1 and 3 in Figure 1 and and Section 3.2. See Figure 4 for a graphical illustration and Appendix B.2 for intuitions. Of equal importance to us is the negations: anti-Lipscthizness and anti-co-Lipschitzness (Definition B.6, B.7) in Appendix B.2. See also Appendix B.2 for the relation between co-Lipschitzness and quasi-isometry in geometric group theory. These concepts are used in Theorem 3.8 and their relations to OOD detection are discussed in Section 3.2.

### 3.1.3 PROVABLE OOD DETECTION PERFORMANCE GUARANTEE FOR VAEs

Our main theoretical result quantifies how well VAEs' minimal sufficient statistics can detect $P_{\mathrm{OOD}}$. At a high level, three major factors capture the hardness of an OOD detection problem. The first is the *dataset property*, such as $m_{\mathrm{inter}}$ (Definition 3.4). The second class is the *function analytic properties* including Lipschitzness and co-Lipschitzness in Section 3.1.2. We introduce the last one, *statistical generalization properties*, which is reflected as test time reconstruction error for the VAEs:

**Definition 3.7** (IID reconstruction distance as intra-distribution margin). The intra-distribution margin, $m_{\mathrm{intra}}$, is defined as:

$$m_{\mathrm{intra}} := \sup_{\mathbf{x}_{\mathrm{IID}} \sim P_{\mathrm{IID}}} \mathrm{d}(\mathbf{x}_{\mathrm{IID}}, \widehat{\mathbf{x}}_{\mathrm{IID}}) \tag{9}$$

We verify VAEs are sufficiently well trained on $P_{\text{IID}}$ by checking $\|\mathbf{x}_{\text{IID}} - \widehat{\mathbf{x}}_{\text{IID}}\|_2$ via sampling from $P_{\text{IID}}$ in test time. Even with our small DC-VAE models, the reconstruction errors are very small (Appendix B.3). We therefore assume $m_{\text{intra}} < \frac{1}{2} m_{\text{inter}}$ henceforth, for any reasonable desired level of separation $m_{\text{inter}}$. Our main theoretical result:

**Theorem 3.8** (Provable OOD detection). *Fix $P_{IID}$, $P_{OOD}$, $m_{intra} > 0$ and $m_{inter} > 2 \cdot m_{intra}$. Assume without loss of generality the corresponding $\arg\min$ in Definition 3.4 for $m_{inter}$ exists, denoted as: $(\epsilon_{IID}^*, \epsilon_{OOD}^*)$. Suppose the encoder $q_\phi : \mathbf{x} \longrightarrow (\mu_{\mathbf{z}}(\mathbf{x}), \sigma_{\mathbf{z}}(\mathbf{x}))$ is co-Lipschitz with degrees $(K, k)$, or the decoder $p_\theta : \mathbf{z} \longrightarrow (\mu_{\mathbf{x}}(\mathbf{z}), \sigma_{\mathbf{x}}(\mathbf{z}))$ is $L$ Lipschitz with $L \leq K$ [8].*

*Then for any metric in the input space $\mathrm{d}_X(\cdot, \cdot)$ [9] upon which $m_{inter}$ and $m_{intra}$ margins are defined, with probability $\geq 1 - (\epsilon_{IID}^* + \epsilon_{OOD}^*)$ over $(P_{IID}, P_{OOD})$, at least one of the following holds:*

$$\|\mu_{\mathbf{z}}(\mathbf{x}_{IID}) - \mu_{\mathbf{z}}(\mathbf{x}_{OOD})\|_2 \geq \frac{m_{inter} - k}{K} \quad and \quad \|\sigma_{\mathbf{z}}(\mathbf{x}_{IID}) - \sigma_{\mathbf{z}}(\mathbf{x}_{OOD})\|_2 \geq \frac{m_{inter} - k}{K} \quad (10)$$

$$\mathrm{d}_X(\mathbf{x}_{OOD}, \widehat{\mathbf{x}}_{OOD}) \geq \frac{2K - L}{2K} m_{inter} - m_{intra} + \frac{kL}{2K} \quad (11)$$

See Appendix B.3 for the proof and Figure 1 for an illustration. These two bounds decouple the minimal sufficient statistics' detection efficacy to: $m_{\text{inter}}$, the desired level of separation (depending on $P_{\text{IID}}$ and $P_{\text{OOD}}$ but independent of models), $L$, the Lipschitz constant of $p_\theta$, $(K, k)$, the co-Lipschitz degrees of $q_\phi$, and $m_{\text{intra}}$, the test time reconstruction errors in $P_{\text{IID}}$. Theorem 3.8 suggests OOD samples can be detected either via the latent code distances (Equation 37) or the reconstruction error (Equation 38). We discuss how Theorem 3.8 is a weaker solution concept than aiming for better $p_\theta$ estimation for OOD detection, its implication on algorithmic design (break VAEs in the right way), its statistical aspects, and its limitations (e.g. hard to track $k$, $K$, $L$ exactly, similar to Lipschitz constants in optimization theory Bubeck et al. (2015)) in Appendix B.3.

## 3.2 NOT ALL OOD SAMPLES ARE CREATED EQUAL, NOT ALL STATISTICS ARE APPLIED THE SAME

This section presents our computation-ready summary statistics. While Equation 38 is readily available, Equation 37 does not manifest itself as computationally friendly, as we need to sample from $P_{\text{IID}}$ in inference time. In this section, we delve further into the geometric and combinatorial structures in VAEs, seeking computationally fast substitutes for Equation 37.

**Not all OOD samples are created equal**: classify $\mathbf{x}_{\text{OOD}}$' likelihood paths to four cases, based on Theorem 3.8, and demonstrated in Figure 1 and 2. Breaking it down this way clarifies how Theorem 3.8 works. We use Definitions 3.5, 3.6, B.6 and B.7 throughout. We set $\mathbf{z} = \mu_{\mathbf{z}}(\mathbf{x})$ (and thus ignore $\sigma_{\mathbf{z}}(\mathbf{x})$) to simplify the notations. The reasoning for $\sigma_{\mathbf{z}}(\mathbf{x})$ is identical and thus omitted.

**Case (1) [$q_\phi$ "many-to-one" and $p_\theta$ reconstructs well: difficult case]**: Corresponding to Figure 1, encoder $q_\phi$ maps both $\mathbf{x}_{\text{OOD}}$ (left yellow 1) and $\mathbf{x}_{\text{IID}}$ (left blue) to nearby regions: $\mathbf{z}_{\text{OOD}} \approx \mathbf{z}_{\text{IID}}$. Furthermore, the decoder $p_\theta$ "tears" $\mathbf{z}_{\text{OOD}}$ nearby regions (middle yellow 1 inside middle blue) to reconstruct both $\mathbf{x}_{\text{OOD}}$ and $\mathbf{x}_{\text{IID}}$ well (right blue and right yellow 1), mapping nearby latent codes to drastically different locations in the visible space. **Case (2) [$q_\phi$ "one-to-one" and $p_\theta$ reconstructs well on $P_{\text{OOD}}$]**: In this scenario, $q_\phi$ maps $\mathbf{x}_{\text{OOD}}$ and $\mathbf{x}_{\text{IID}}$ to different latent locations. As long as $\mathbf{x}_{\text{OOD}}$ is far from $\mathbf{x}_{\text{IID}}$ in the visible space, $\mathbf{z}_{\text{OOD}}$ is far from any $\mathbf{z}_{\text{IID}}$, but $\mathbf{x}_{\text{OOD}}$ is well reconstructed. The statistics $\|\mathbf{z}_{\text{IID}} - \mathbf{z}_{\text{OOD}}\|_2$ can flag $\mathbf{x}_{\text{OOD}}$. **Case (3) [ $q_\phi$ "many-to-one" and $p_\theta$ reconstructs $P_{\text{OOD}}$ poorly ]**: Like Case (1), $q_\phi$ makes "many-to-one" errors: $\mathbf{z}_{\text{IID}} \approx \mathbf{z}_{\text{OOD}}$ for some $\mathbf{x}_{\text{IID}}$. But thanks to $p_\theta$'s continuity, $\widehat{\mathbf{x}}_{\text{OOD}}(\mathbf{z}_{\text{OOD}}) \approx \widehat{\mathbf{x}}_{\text{IID}}(\mathbf{z}_{\text{IID}})$. If $\mathbf{x}_{\text{OOD}}$ is away from $\mathbf{x}_{\text{IID}}$ by a detectable margin, and VAEs are well trained: $\widehat{\mathbf{x}}_{\text{IID}} \approx \mathbf{x}_{\text{IID}}$, $\|\mathbf{x}_{\text{OOD}} - \widehat{\mathbf{x}}_{\text{OOD}}\|_2 \approx \|\mathbf{x}_{\text{OOD}} - \widehat{\mathbf{x}}_{\text{IID}}\|_2 \approx \|\mathbf{x}_{\text{OOD}} - \mathbf{x}_{\text{OOD}}\|_2$ is large. **Case (4) [ $q_\phi$ "one-to-one" and $p_\theta$ reconstructs $P_{\text{OOD}}$ poorly ]**: When both Case (2) and Case (3) are true, it is detectable either way.

**Not all statistics are sufficient and simple: empirical concentrations and distance to $\mathbf{z}_{\text{IID}}$ latent manifold**. Previous discussion leaves out the calculation of Equation 37. Because this involves

---

[8]This condition is evoked when $q_\phi$ fails to be co-Lipschitz with degrees $(K, k)$. $L \leq K$ is sensible because VAEs learns to reconstruct $P_{\text{IID}}$.

[9]We mean metric spaces that obey the triangle inequality. This is extremely general, including widely used $l^\infty$ in adversarial robustness, perceptual distance in vision Gatys et al. (2016), etc. Our result also extends to any metric in the latent spaces. We use $l^2$ norm for the latent variable parameters for simplicity.

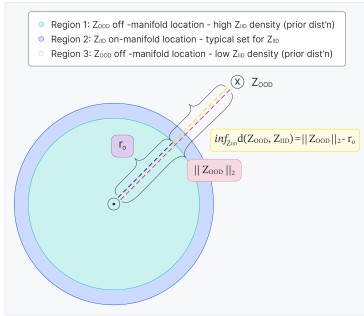

Figure 5: **Illustration of $v$ statistics in Equation 12.** Region 1 (turquoise) and Region 3 (grey) indicate OOD regions, Region 2 (blue) IID is for latent manifold region. $\mu_{\mathbf{z}}(\mathbf{x})$ empirically concentrates around a spherical shell. To screen $\mathbf{x}_{\text{OOD}}$, we can track $\mathbf{z}_{\text{OOD}} := \mu_{\mathbf{z}}(\mathbf{x}_{\text{OOD}})$, and compute its distances to the IID latent manifold, $\inf_{\mathbf{z}_{\text{IID}}} \mathrm{d}(\mathbf{z}_{\text{IID}}, \mathbf{z}_{\text{OOD}})$. Since $\mathbf{z}_{\text{IID}}$ concentrates on some spherical shell of radius $r_0$, $\inf_{\mathbf{z}_{\text{IID}}} \mathrm{d}(\mathbf{z}_{\text{IID}}, \mathbf{z}_{\text{OOD}})$ can be efficiently approximated. This is one illustrative case, our reasoning holds even if $\mathbf{z}_{\text{OOD}}$ is in the blue or turquise region.

| **Algorithm**: Two Stage OOD Training | **Algorithm**: Dual Feature Levels OOD Detection |
|---|---|
| 1: Input: $\mathbf{x} \in \mathrm{D}_{\text{train}}$; | 1: Input: $\mathbf{x} \sim \mathbb{P}_{\text{OOD}}$; |
| 2: Train VAE for $\mathrm{D}_{\text{train}}$; | 2: Compute $(u(\mathbf{x}), v(\mathbf{x})), w(\mathbf{x})$ |
| 3: Compute $(u(\mathbf{x}), v(\mathbf{x})), w(\mathbf{x})$ (Eq. 12) for the trained VAE; | (Eq. 12) for the trained VAE; |
| 4: Use $(u(\mathbf{x}), v(\mathbf{x})), w(\mathbf{x})$ in the second stage training, as input data to fit COPOD; | 3: Use the fitted COPOD, $D$ to get a decision score $D(\mathbf{x})$; |
| 5: Output: fitted COPOD on $(u(\mathbf{x}), v(\mathbf{x})), w(\mathbf{x})$ in training dataset, $\mathrm{D}_{\text{train}}$ $D(\mathbf{x})$ to $\{D(\mathbf{x})\}_{\mathbf{x} \in \mathrm{D}_{\text{train}}}$ | 4: Output: Determine if $\mathbf{x}$ is OOD by comparing $D(\mathbf{x})$ to $\{D(\mathbf{x})\}_{\mathbf{x} \in \mathrm{D}_{\text{train}}}$ |

sampling from $P_{\text{IID}}$ and $P_{\text{OOD}}$, it appears non-trivial to compute. We propose an approximation based on the empirical observation that $\mu_{\mathbf{z}}(\mathbf{x}_{\text{IID}})$ concentrates around the spherical shell, $\mathcal{S}_{\mu(\mathbf{z})}$ centered at $\mathbf{0}$ with radius $r_0$. (Figure 2). In other words, the supports of $\mu_{\mathbf{z}}(\mathbf{x}_{\text{IID}})$ where $\mathbf{x}_{\text{IID}} \sim P_{\text{IID}}$, can be approximated by a spherical shell. Suppose the (unknown but fixed) spherical radius is $r_0$. For any $\mathbf{x}_{\text{OOD}}$ and most $\mathbf{x}_{\text{IID}}$, $\|\mu_{\mathbf{z}}(\mathbf{x}_{\text{IID}}) - \mu_{\mathbf{z}}(\mathbf{x}_{\text{OOD}})\| \approx |\|\mu_{\mathbf{z}}(\mathbf{x}_{\text{OOD}})\| - r_0|$. The argument for $\sigma_{\mathbf{z}}$ is identical and won't be repeated. A formalization of the aforementioned heuristics is given in Appendix B.4.1. We therefore further modify the training objective to encourage this concentration effect. The details of our modification can be found in Appendix B.5.1. We hence finalize the OOD scoring statistics:

$$u(\mathbf{x}) = \|\mathbf{x} - \widehat{\mathbf{x}}\|_2 = \|\mathbf{x} - \mu_{\mathbf{x}}(\mu_{\mathbf{z}}(\mathbf{x}))\|_2 \tag{12}$$

$$v(\mathbf{x}) = \|\mu_{\mathbf{z}}(\mathbf{x})\|_2 \approx |\|\mu_{\mathbf{z}}(\mathbf{x})\|_2 - r_0| \tag{13}$$

$$w(\mathbf{x}) = \|\sigma_{\mathbf{z}}(\mathbf{x})\|_2 \approx |\|\sigma_{\mathbf{z}}(\mathbf{x})\|_2 - r_I| \tag{14}$$

where $r_I$ (or $r_I$ for $\sigma_{\mathbf{z}}$) is dropped because: (1) the operation $|\cdot - r_0|$ (or $|\cdot - r_I|$) is a function of $\|\mu_{\mathbf{z}}(\mathbf{x})\|_2$ (or $\|\sigma_{\mathbf{z}}(\mathbf{x})\|_2$), so it does not contain more information [10]; (2) it saves us from estimating $r_0$ (or $r_I$). These simple functions of the minimal sufficient statistics align with the geometry of Theorem 3.8 while being computationally fast. They also enjoy provable guarantees, shown in Appendix B.4.1. Theorem 3.8 also has implications on algorithmic design, and we explore such heuristics in Appendix B.4.2. Section 4 details how our theory and heuristics translate to OOD detection algorithms.

## 4 METHODOLOGY AND ALGORITHM

In this section, we describe our two-stage algorithm, with a similar framework as Morningstar et al. (2021). Our algorithm can be used for only one VAE model (LPath-1M) or a pair of two models (LPath-2M). In the first stage (**neural feature extraction**), for LPath-2M, we train two VAEs . One VAE has a very high latent dimension (e.g. 1000) and another with a very low dimension (e.g. 1 or 2), following our analysis in Section B.4.2 and B.4.3. In the second stage (**classical density estimation**), we extract the following statistics, $(u(\mathbf{x})_{\text{low D}}, v(\mathbf{x})_{\text{high D}}, w(\mathbf{x})_{\text{high D}})$ as in Equations 12, where $u(\mathbf{x})_{\text{low D}}$ is taken from the low dimensional VAE and $v(\mathbf{x})_{\text{high D}}, w(\mathbf{x})_{\text{high D}}$ from the high dimensional VAE. Section B.4.3 explains the reasoning behind such combination. For LPath-1M, we use the same VAE to extract all of $u(\mathbf{x}), v(\mathbf{x}), w(\mathbf{x})$. We then fit a classical statistical density

---

[10]Ddata processing inequality from information theory is one way to formalize this: since $\mathbf{X} \to \|\mu_{\mathbf{z}}(\mathbf{X})\|_2 \to |\|\mu_{\mathbf{z}}(\mathbf{X})\|_2 - r_0|$ forms a Markov chain, the following mutual information inequality holds: $I(\mathbf{X}; \|\mu_{\mathbf{z}}(\mathbf{X})\|_2) \geq I(\mathbf{X}; |\|\mu_{\mathbf{z}}(\mathbf{X})\|_2 - r_0|)$. Our theoretical discussion around $\mu_{\mathbf{z}}(\mathbf{X})$'s concentration suggests $|\|\mu_{\mathbf{z}}(\mathbf{X})\|_2 - r_0|$, but data processing inequality gives us a computationally faster and no less informative candidate $\|\mu_{\mathbf{z}}(\mathbf{X})\|_2$.

| IID
OOD | CIFAR10 | | | | SVHN | | | FMNIST | | | MNIST | | |
|---|---|---|---|---|---|---|---|---|---|---|---|---|---|
| | SVHN | CIFAR100 | Hflip | Vflip | CIAFR10 | Hflip | Vflip | MNIST | Hflip | Vflip | FMNIST | Hflip | Vflip |
| ELBO | 0.08 | 0.54 | 0.5 | 0.56 | **0.99** | 0.5 | 0.5 | 0.87 | 0.63 | 0.83 | **1.00** | 0.59 | 0.6 |
| LR (Xiao et al., 2020) | 0.88 | N/A | N/A | N/A | 0.92 | N/A | N/A | 0.99 | N/A | N/A | N/A | N/A | N/A |
| BIVA (Havtorn et al., 2021) | 0.89 | N/A | N/A | N/A | **0.99** | N/A | N/A | 0.98 | N/A | N/A | **1.00** | N/A | N/A |
| DoSE (Morningstar et al., 2021) | 0.97 | 0.57 | 0.51 | 0.53 | **0.99** | 0.52 | 0.51 | **1.00** | 0.66 | 0.75 | **1.00** | **0.81** | 0.83 |
| Fisher (Bergamin et al., 2022) | 0.87 | 0.59 | N/A | N/A | N/A | N/A | N/A | 0.96 | N/A | N/A | N/A | N/A | N/A |
| DDPM (Liu et al., 2023) | 0.98 | N/A | 0.51 | 0.63 | **0.99** | **0.62** | **0.58** | 0.97 | 0.65 | **0.89** | N/A | N/A | N/A |
| LMD (Graham et al., 2023) | **0.99** | 0.61 | N/A | N/A | 0.91 | N/A | N/A | 0.99 | N/A | N/A | **1.00** | N/A | N/A |
| LPath-1M-COPOD (Ours) | **0.99** | **0.62** | **0.53** | 0.61 | **0.99** | 0.55 | 0.56 | **1.00** | 0.65 | 0.81 | **1.00** | 0.65 | **0.87** |
| LPath-2M-COPOD (Ours) | 0.98 | **0.62** | **0.53** | **0.65** | 0.96 | 0.56 | 0.55 | 0.95 | **0.67** | 0.87 | **1.00** | 0.77 | 0.78 |
| LPath-1M-MD (Ours) | **0.99** | 0.58 | 0.52 | 0.60 | 0.95 | 0.52 | 0.52 | 0.97 | 0.63 | 0.82 | **1.00** | 0.75 | 0.76 |

Table 1: AUROC of OOD Detection with different IID and OOD datasets. LPath-1M is LPath with one model, LPath-2M is LPath with two models, one VAE with overly small latent space and another with overly large latent space.

estimation algorithm (COPOD Li et al. (2020) or MD Lee et al. (2018); Maciejewski et al. (2022)) to $(u(\mathbf{x}), v(\mathbf{x}), w(\mathbf{x}))$ for LPath-1M or $(u(\mathbf{x})_{\text{low D}}, v(\mathbf{x})_{\text{high D}}, w(\mathbf{x})_{\text{high D}})$ for LPath-2M viewed as second stage training data. This second stage scoring is our OOD decision rule, detecting OOD according to Theorem 3.8.

## 5 EXPERIMENTS

We compare our methods with state-of-the-art OOD detection methods Kirichenko et al. (2020); Xiao et al. (2020); Havtorn et al. (2021); Morningstar et al. (2021); Bergamin et al. (2022); Liu et al. (2023); Graham et al. (2023), under the unsupervised, single batch, no data inductive bias assumption setting. Following the convention in those methods, we have conducted experiments with a number of common benchmarks, including CIFAR10 (Krizhevsky & Hinton, 2009), SVHN (Netzer et al., 2011), CIFAR100 (Krizhevsky & Hinton, 2009), MNIST(LeCun et al., 1998), FashionMNIST (FMNIST)(Xiao et al., 2017), and their horizontally flipped and vertically flipped variants.

**Experimental Results** in Table 1, shows that our methods surpass or are on par with state-of-the-art (SOTA). Because our setting assumed no access to labels, batches of test data, or even any inductive bias on the dataset, OOD datasets like Hflip and VFlip become very challenging (reflected as small $m_{\text{inter}}$). Most prior methods achieved only near chance AUROC on Vflip and Hflip for CIFAR10 and SVHN as IID data. This is expected, because horizontally flipped CIFAR10 or SVHN differs from in-distribution only by one latent dimension. Even so, our methods still managed to surpass prior SOTA in some cases, though only marginally. This improvement is made more significant given that that we only used a very small VAE architecture, while competitive prior methods used larger models like Glow (Kingma & Dhariwal, 2018) or diffusion models (Rombach et al., 2022). We remark that ours clearly exceed other VAEs based methods Xiao et al. (2020); Havtorn et al. (2021), and is the only VAE based method that is competitive against bigger models. More experimental details, including various ablation studies are in Appendix C, D.

**Minimality and sufficiency are advantageous**. DoSE Morningstar et al. (2021) conducted experiments on VAEs with five carefully chosen statistics. Assuming better results are reported therein, our methods surpass their Glow based scores, which should in turn be better than their VAEs'. On one hand, Glow's likelihood is arguably much better estimated than our small DC-VAE model, by comparing the generative samples' quality. On the other hand, their statistics appear to be more sophisticated. However, our simple method based on LPath manages to surpass their scores. This showcases the benefits of minimal sufficient statistics.

## 6 CONCLUSION

We presented the likelihood path principle applied to unsupervised, one-sample OOD detection. This leads to our provable method, which is arguably more interesting as OOD data are unknown unknowns. Our theory and methods are supported by SOTA results. In future works, we plan to adapt our principles and techniques to more powerful DGMs, such as Glow or Diffusion models.

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

# A   APPENDIX FOR SECTIION 1

## A.1   RELATED WORK

Prior works have approached OOD detection from various perspectives and with different data assumptions, e.g., with or without access to training labels, batches of test data or single test data point in a steaming fashion, and with or without knowledge and inductive bias of the data. In the following, we give an overview organized by different data assumptions with a focus on where our method fits.

The first assumption is whether the method has access to training labels. There has been extensive work on classifier-based methods that assume access to training labels (Hendrycks & Gimpel, 2016; Frosst et al., 2019; Sastry & Oore, 2020; Bahri et al., 2021; Papernot & McDaniel, 2018; Osawa et al., 2019; Guénais et al., 2020; Lakshminarayanan et al., 2016; Pearce et al., 2020). Within this category of methods, there are different assumptions as well, such as access to a pretrained-net, or knowledge of OOD test examples. See Table 1 of (Sastry & Oore, 2020) for a summary of such methods.

When we do not assume access to the training labels, this problem becomes a more general one and also harder. Under this category, some methods assume access to a batch of test data where either all the data points are OOD or not (Nalisnick et al., 2019). A more general setting does not assume OOD data would come in as batches. Under this setup, there are methods that implicitly assume prior knowledge of the data, such as the input complexity method (Serrà et al., 2019), where the use of image compressors implicitly assumed an image-like structure, or the likelihood ratio method (Ren et al., 2019) where a noisy background model is trained with the assumption of a background-object structure.

Lastly, as mentioned in Section 1, our method is among the most general and difficult setting where we assumed no access to labels, batches of test data, or even any inductive bias of the dataset (Xiao et al., 2020; Kirichenko et al., 2020; Havtorn et al., 2021; Ahmadian & Lindsten, 2021; Morningstar et al., 2021; Bergamin et al., 2022). Xiao et al. (2020) fine-tune the VAE encoders on the test data and take the likelihood ratio to be the OOD score. Kirichenko et al. (2020) trained RealNVP on EfficientNet (Tan & Le, 2020) embeddings and use log-likelihood directly as the OOD score. Havtorn et al. (2021) trained hierarchical VAEs such as HVAE and BIVA and used the log-likelihood directly as the OOD score. Recent works by Morningstar et al. (2021); Bergamin et al. (2022); Liu et al. (2023); Graham et al. (2023) were discussed in Section 1. Notably, recent works benefit from bigger models such as Glow Morningstar et al. (2021) or diffusion models Liu et al. (2023); Graham et al. (2023). We compare our method with the above methods in Table 1.

# B   APPENDIX FOR SECTION 3

**Definition of diameter in metric-measure spaces**

We define $\mathrm{Diameter}(U) = \sup_{x,y \in U} \mathrm{d}(x,y)$ (as a generalization of diameter of a rectangle) for a subset $U \subset X$ in a metric-measure space $(X, \mathrm{d}_X, \mu)$.

## B.1   SUPPLEMENTARY MATERIALS FOR SECTION 3.1.1

**Definition of supports in metric-measure spaces**

**Definition B.1.** Let $(X, \mathrm{d}_X, \mu)$ be a metric-measure space. The support of the measure $\mu$ is the set $\{x \in X \mid \mu(B_{\mathrm{d}_X}(x,r)) > 0, \text{ for all } r > 0\}$ where $B_{\mathrm{d}_X}(x,r) = B_r(x)$ denotes the metric ball with center at $x$ and radius $r$. The latter abbreviated notation is used when the context is clear.

In particular, if $X \subset \mathbb{R}^n$, the support is a subset of $\mathbb{R}^n$. For a random variable defined on a metric-measure space, we define the support of its distribution to be the support of the corresponding probability measure.

**More examples to illustrate Definition 3.2** In this section, we give more examples to illustrate Definition 3.2.

*Example* B.2 (Partially overlapping Gaussians). To see how nearly essential support can be useful, consider Gaussian $\mathcal{N}(0,1)$. Although $\mathrm{Supt}(\mathcal{N}(0,1)) = \mathbb{R}$, the interval $[-3,3]$ contain 99.7 % of the events. $[-3,3]$ is a nearly essential support for $\mathcal{N}(0,1)$ with $\epsilon = 0.003$.

*Example* B.3 (Essential distance between Gaussians). To concretize essential distance between distributions, we can consider $P_{\mathrm{IID}} = \mathcal{N}(-6,1)$ and $P_{\mathrm{OOD}} = \mathcal{N}(6,1)$. Because both have supports $\mathbb{R}$, $\mathrm{d}(\mathrm{Supt}(P_{\mathrm{IID}}), \mathrm{Supt}(P_{\mathrm{OOD}})) = 0$. However, their samples are fairly separable. If we choose $\epsilon = 0.003$ to truncate two tails symmetrically for both, $\mathrm{D}_{X|\epsilon_{\mathrm{IID}}, \epsilon_{\mathrm{OOD}}}(P_{\mathrm{IID}}, P_{\mathrm{OOD}}) = \mathrm{D}_{\mathbb{R}|0.003,0.003}(\mathcal{N}(-6,1), \mathcal{N}(6,1)) = 6$.

*Example* B.4 (Partially overlapping uniforms). Consider $\mathcal{U}_{\mathrm{IID}}$ supported on $[0,1]$ and $\mathcal{U}_{\mathrm{OOD}}$ supported on $[0.75, 1.75]$. Let $m_{\mathrm{inter}} = 0.25$. Then setting $\epsilon^*_{\mathrm{IID}} = \epsilon^*_{\mathrm{OOD}} = 0.25$, and ignoring parts of $\mathcal{U}_{\mathrm{IID}}$ and $\mathcal{U}_{\mathrm{OOD}}$, we have: $\mathrm{ESupt}(\mathcal{U}_{\mathrm{IID}}; -0.25) = [0, 0.75]$, $\mathrm{ESupt}(\mathcal{U}_{\mathrm{OOD}}; -0.25) = [1, 1.75]$. $\mathrm{D}_{X|m_{\mathrm{inter}}=0.25}(\mathcal{U}_{\mathrm{IID}}, \mathcal{U}_{\mathrm{OOD}}) = \mathrm{D}_{X|\epsilon^*_{\mathrm{IID}}=0.25, \epsilon^*_{\mathrm{OOD}}=0.25}(\mathcal{U}_{\mathrm{IID}}, \mathcal{U}_{\mathrm{OOD}}) = 0.25$. In other words, with probability at least 1 - (0.25 + 0.25) = 0.5 over the joint distribution $\mathcal{U}_{\mathrm{IID}}, \mathcal{U}_{\mathrm{OOD}}, \mathcal{U}_{\mathrm{IID}}, \mathcal{U}_{\mathrm{OOD}}$ are separated by 0.25.

*Example* B.5 (Totally overlapped uniforms). Consider $\mathcal{U}_{\mathrm{IID}} = \mathcal{U}_{\mathrm{OOD}}$ supported on $[0,1]$. Let $m_{\mathrm{inter}} = 0$. Then setting $\epsilon^*_{\mathrm{OOD}} = \epsilon^*_{\mathrm{OOD}} = 0.5$, we have: $\mathrm{ESupt}(\mathcal{U}_{\mathrm{IID}}; -0.5) = [0, 0.5]$, $\mathrm{ESupt}(\mathcal{U}_{\mathrm{OOD}}; -0.5) = [0.5, 1]$. $\mathrm{D}_{X|m_{\mathrm{inter}}=0.5}(\mathcal{U}_{\mathrm{IID}}, \mathcal{U}_{\mathrm{OOD}}) = \mathrm{D}_{X|\epsilon^*_{\mathrm{IID}}=0.5, \epsilon^*_{\mathrm{OOD}}=0.5}(\mathcal{U}_{\mathrm{IID}}, \mathcal{U}_{\mathrm{OOD}}) = 0$. In other words, with probability at least 1 - (0.5 + 0.5) = 0 over the joint distribution $\mathcal{U}_{\mathrm{IID}}, \mathcal{U}_{\mathrm{OOD}}, \mathcal{U}_{\mathrm{IID}}, \mathcal{U}_{\mathrm{OOD}}$ are separated by 0. This example shows that the definition captures the extreme case well and remain well-behaved.

**Essential separation in measure theoretic terms**

Definitions 3.1, 3.2, 3.3 are related to some standard constructions in measure theory. Our main exposition does not assume readers are familiar with measure theory, in order to make the main paper more accessible. Moreover, our writings are tailored for the applications of interests.

In here, we cover the measure theoretic perspective here for completeness. From the mathematical perspective, only the inter-dependency between the probabilistic notions ($\epsilon_{\mathrm{IID}}, \epsilon_{\mathrm{OOD}}$), and the margin or distance $m_{inter}$ between $P_{\mathrm{IID}}$ and $P_{\mathrm{OOD}}$ are new constructions. Definition 3.1 can be rephrased in the following way:

In the spirits of measure decomposition, we divide $P_{\mathrm{IID}}$ and $P_{\mathrm{OOD}}$ into components $P_{\mathrm{IID}} = P_{\mathrm{IID}}^{\mathrm{likely}} + P_{\mathrm{IID}}^{\mathrm{unlikely}}$ and $P_{\mathrm{OOD}} = P_{\mathrm{OOD}}^{\mathrm{likely}} + P_{\mathrm{OOD}}^{\mathrm{unlikely}}$ such that:

- $P_{\mathrm{IID}}^{\mathrm{likely}} \perp P_{\mathrm{IID}}^{\mathrm{unlikely}}$

- $P_{\mathrm{OOD}}^{\mathrm{likely}} \perp P_{\mathrm{OOD}}^{\mathrm{unlikely}}$

- $P_{\mathrm{IID}}^{\mathrm{likely}} \perp P_{\mathrm{OOD}}^{\mathrm{likely}}$

- $P_{\mathrm{IID}}^{\mathrm{unlikely}} \leq \epsilon_{\mathrm{IID}}$

- $P_{\mathrm{OOD}}^{\mathrm{unlikely}} \leq \epsilon_{\mathrm{OOD}}$

where the notation $\perp$ means the two measures are supported on disjoint sets. These are reminiscent to the Lebesgue decomposition theorem.

The more mathematical readers may notice that our constructions, Definitions 3.1, 3.2, 3.3 are based on $\min$, $\max$, $\inf$ and $\sup$ operators. These are intuitively related to Hausdorff distances, and more generally min-max theory applied to geometry (Chapter 2 of Chu & Li (2023)). While exploring and further extending our constructions is interesting, it is beyond the scope of the present paper and is left to future works.

### B.2 SUPPLEMENTARY MATERIALS FOR SECTION 3.1.2

In this section, we give the negations of Lipschitzness and Definition 3.6.

**Definition B.6** (Anti-Co-Lipschitz: region-wise "many-to-one"). Let $K > 0, k \geq 0$ be given. Under the same settings as Definition 3.5, a function $f : X \to Y$ is **anti-co-Lipschitz** with degrees $(K, k)$, if there exist $y \in Y$ and $R > 0$ such that:

$$\mathrm{Diameter}(f^{-1}(B_R(y))) > K \cdot \mathrm{Diameter}(B_R(y)) + k \qquad (15)$$

Heuristically, we call it region-wise "many-to-one", because the diameter of the inverse image, $f^{-1}(B_R(y))$, is bounded below. That means $f^{-1}(B_R(y))$ sweeps out a big region. In other words, these far away points in the domain are mapped by $f$ to a small region in the codomain/target space.

**Definition B.7** (Anti-Lipschitz: region-wise "one-to-many"). Under the same settings as Definition 3.5, a function $f : X \to Y$ is **anti-Lipschitz** with degrees $L$, if there exist $x \in Y$ and $R > 0$ such that:

$$\text{Diameter}(f(B_R(x))) > L \cdot \text{Diameter}(B_R(x)) \tag{16}$$

Heuristically, we call it region-wise "one-to-many", because the diameter of $f(B_R(y))$, is bounded below. That means $f(B_R(x))$ sweeps out a big region in the codomain/target space. In other words, small regions are mapped by $f$ to a big region in the codomain. Intuitively, $L$-Lipschitz continuity quantifies how much $f$ can stretch a metric ball. We call it region-wise not "one-to-many", because Lipschtiz functions cannot stretch one small region into a region with big diameter. As $L \to \infty$, however, $f$ becomes increasingly more "one-to-many", approaching a discontinuous function. More heuristics or interpretations can be found below.

**Equivalence of Definition 3.5 to the standard Lipschitzness.**

*Proof.* Recall the classic definition:

**Definition B.8** (L-Lipschitz). Let $(X, \mathrm{d}_X, \mu)$ and $(Y, \mathrm{d}_Y, \nu)$ be two metric-measure spaces, with equal (probability) measures $\mu(X) = \nu(Y)$. Let $L > 0$ be fixed. A function $f : X \to Y$ is $L$-**Lipschitz**, if for any any $x_1, x_2 \in X$:

$$\mathrm{d}_Y(f(x_1), f(x_2)) \leq L\mathrm{d}_X(x_1, x_2) \tag{17}$$

1. Classic Lipschitz $\to$ geometric Lipschitz.

Take any $x_1, x_2 \in B_R(x_1)$ such that $\mathrm{d}(y_1, y_2) \approx \text{Diameter}(f(B_R(x)))$ and $\mathrm{d}(x_1, x_2) = 2R$ for the corresponding $y_1, y_2$. Without loss of generality and saving us from tracking $\epsilon$, assume $\mathrm{d}(y_1, y_2) = \text{Diameter}(f(B_R(x)))$. By classic Lipschitz condition, $\mathrm{d}(y_1, y_2) \leq L\mathrm{d}(x_1, x_2)$, so: $\mathrm{d}(y_1, y_2) = \text{Diameter}(f(B_R(x))) \leq L \cdot \text{Diameter}(B_R(x)) = 2R$.

2. Geometric Lipschitz $\to$ classic Lipschitz. Take any $x_1, x_2 \in X$ and define $R = \frac{\mathrm{d}(x_1, x_2)}{2}$. By geometric Lipschitzness, $\text{Diameter}(f(B_R(x_1))) \leq L \cdot \text{Diameter}(B_R(x_1)) = 2R = \mathrm{d}(x_1, x_2)$. Since $f(x_1), f(x_2) \in f(B_R(x_1))$, $\mathrm{d}(f(x_1), f(x_2)) \leq L \cdot \mathrm{d}(x_1, x_2)$. □

**More discussion and Intuition for Lipschitz and co-Lipschitz**

We discuss our new definitions with more details. We recall some standard definitions before defining ours. We let $f^{-1}$ denote the pre-image or inverse image of the function $f$. Recall $\text{Diameter}(U)$ is defined as $\sup_{x_1, x_2 \in U} \mathrm{d}_X(x_1, x_2)$ in a metric space $(X, \mathrm{d}_X)$. We remind ourselves that a functions is *injective* or *one-to-one*, if for any $y \in Y$, $f^{-1}(y) = x$, i.e. $f^{-1}(y)$ is a singleton set. Otherwise, a function is many-to-one.

Our key observation is that a generalization of the above characterizes VAEs' abilities to detect OOD samples. We introduce quantitative analogues to capture how one-to-one and many-to-one a function $f$ is. Concretely, a function is one-to-one, if the inverse image of a point is one point. Having only one point in both the domain and codomain can be interpreted as a way of measuring the size of a set. This naturally admits two extensions, by relaxing the sizes of sets in both domain and codomain. For example, we can measure the size of the set $f^{-1}(y)$.

We begin the discussion in the domain. we mostly use $\text{Diameter}(f^{-1}(y))$ as in Landweber et al. (2016). If $\text{Diameter}(f^{-1}(y))$ is big, we can say it is relatively "more" many-to-one. Otherwise, it is "less" many-to-one or more "one-to-one". Consider the encoder map, $q_\phi : \mathbf{x} \longrightarrow (\mu_{\mathbf{z}}(\mathbf{z}), \sigma_{\mathbf{z}}(\mathbf{z}))$. We care about how "one-to-one" $f$ is, because we don't want to to be "many-to-one" as both IID and OOD samples can be mapped to the same latent code neighborhood. Definition 3.6 relaxes injectivity in two ways: 1. taking $R \to 0$ and $k = 0$, Definition 3.6 states that $f^{-1}(y)$ has zero diameter; 2. When $k = 0$, applying co-Lipschitz with non-asymptotic radius $R > 0$, we say $f$ is region-wise "one-to-one" if for "small" $R$, $f^{-1}(B_R(y))$ has "small" diameter.

In machine learning, we seldom care one about latent code, but the continuous neighborhood around it. For this reason, we consider the inverse image of a metric ball around a point. Instead of measuring $\text{Diameter}(f^{-1}(y))$, we thus measure: $\text{Diameter}(f^{-1}(B_R(y)))$. This quantifies how "one-to-one" $f$ is: if the inverse image of a metric ball in the codomain has small diameters, we then say $f$ is region-wise "one-to-one". Otherwise, it is very "many-to-one":

To gain some intuition on Definition B.6, if $(K = 100, k = 0)$, $f = q_\phi$ can map two points more than $100R$ away to the same latent code. If such one point happens to be OOD and another is IID, we won't be able to detect the OOD in the latent space. On the other hand, if $f = q_\phi$ is region-wise one-to-one or co-Lipschitz at $(K = 100, k = 0)$ and $\mathbf{x}_{\text{OOD}}$ is 100 distance away from $\mathbf{x}_{\text{IID}}$, we can detect it in theory.

Definition B.6 and Definition 3.6 form a natural pair. They are kind of the opposite of the other. Note that they are both defined in the backward manner: both are defined in the codomain using inverse images. That is, we compare the diameters between: $f^{-1}(B_R(y))$ in the domain and $B_R(y)$ in the codomain. We now discuss the next pair.

Note that $f(B_R(x))$ is in the codomain and $B_R(x)$ is in the domain in Definition 3.5 and Definition B.7. These are in the forward direction. They form a polar pair, just like region-wise one-to-one and region-wise many-to-one. We end this discussion by the next lemma, which formalizes in a sense L-Lipschiz maps cannot be "one-to-many".

**Lemma B.9** ($L$-Lipschitz functions cannot be one-to-many with degree $L$). *Let $f : \mathbf{z} \in \mathbb{R}^m \longrightarrow \mathbb{R}^n$ be a $L$-Lipschitz function. Then $f$ cannot be one-to-many with degree larger than $L$.*

The proof directly follows from definition and we omit it.

**Co-Lipschitzness and Quasi-Isometric Embedding**

The high level idea behind Sections 3.1.1 and 3.1.2 is to seek relaxed or "soft" versions of the "hard" versions of classical metric space separations, distances, and Lipscthitzness. The constructions are inspired by the field of quantitative geometry and topology. Concretely, our definition of co-Lipschtizness, Definition, 3.6 is closely related to quasi-isometry in geometric group theory. We now relate them rigorously.

The high level idea of quasi-isomety is to relax the more rigid concept of isometry by an affine transformation type of inequalities. Recall the definition of isometry:

**Definition B.10** (Isometry). Let $(M_1, \text{d}_1)$ and $(M_2, \text{d}_2)$ be metric spaces. A map $f : M_1 \to M_2$ is called an isometry or distance preserving if for any $x, y \in M_1$, we have:

$$\text{d}_1(x, y) = \text{d}_2(f(x), f(y))$$

The requirement seems a little too rigid to be useful in probabilistic applications, for example, when models are learnt approximately, or that they differ by some scales locally. The next relaxed one, allowing more room (at the linear rate) for errors, is arguably more suitable.

**Definition B.11** (Quasi-Isometry). Suppose $f : M_1 \to M_2$ between two metric spaces as in Definition B.10. Then $f$ is called a quasi-isometry from $(M_1, \text{d}_1)$ to $(M_2, \text{d}_2)$ if there exist constants $A \geq 1, B \geq 0$, and $C \geq 0$ such that the following two properties both hold:

1. For every two points $x$ and $y$ in $M_1$, the distance between their images is up to the additive constant $B$ within a factor of $A$ of their original distance:

$$\forall x, y \in M_1 : \frac{1}{A}\text{d}_1(x, y) - B \leq \text{d}_2(f(x), f(y)) \leq A\text{d}_1(x, y) + B \tag{18}$$

2. Every point of $M_2$ is within the constant distance $C$ of an image point. More formally:

$$\forall z \in M_2 : \exists x \in M_1 : \text{d}_2(z, f(x)) \leq C.$$

The two metric spaces $(M_1, \text{d}_1)$ and $(M_2, \text{d}_2)$ are called **quasi-isometric** if there exists a quasi-isometry $f$ from $(M_1, \text{d}_1)$ to $(M_2, \text{d}_2)$. A map is called a **quasi-isometric embedding** if it satisfies the first condition but not necessarily the second. In other words, $(M_1, \text{d}_1)$ is quasi-isometric to a subspace of $(M_2, \text{d}_2)$.

Quasi-isometric embedding is a much more relaxed concept, because we allow the additional scale parameters $A, B$ that control how $M_1$ and $M_2$ look alike, at scales $A, B$.

The right hand side of Equation 18 generalizes Lipschitzness by an additional additive constant $B$. It is known as *coarse-Lipscthiz* (Proposition 2.2 in Lancien & Dalet (2017)). At first glance, co-Lipschitzness (Definition 3.6) is defined in terms diameter of the pre-image or inverse-image $f^{-1}$, and may not be readily related to quasi-isometry. In the geometric spirit, Definition 3.6 should be called co-coarse-Lipschiz, and Theorem 3.8 is perhaps better framed under coarse-Lipschitz and co-coarse-Lipschiz settings. But since these terms are less widely used in machine learning, our exposition in the main paper does not delve into the mathemtical fine differences. We now relate the left hand side of Equation 18 to co-Lipschitzness.

**Lemma B.12** (Equivalance of LHS of Equation 18 and co-Lipschitizness). *Under the same settings as Definition B.11, the left hand side of Equation 18,*

$$\forall x, y \in M_1 : \mathrm{d}_1(x, y) \leq K \mathrm{d}_2(f(x), f(y)) + k \tag{19}$$

*is equivalent to Definition 3.6 up to a constant factor of 2, i.e. For any $y \in Y$, any $R > 0$:*

$$\mathrm{Diameter}(f^{-1}(B_R(y))) \leq K \cdot \mathrm{Diameter}(B_R(y)) + k \tag{20}$$

The significance of this lemma is that it opens up doors on how we may empirically measure or even certify the co-Lipschitzness. Co-Lipschtizness is defined in terms of the diameter of the inverse image, which can be very difficult to estimate. Now, the lemma suggests measuring encoder's co-Lipschitz degrees by means of encoder's "bi-Lipschitz" alike constants. This in turn allows us to apply techniques from related fields, such as lower bound on adversarial perturbations Peck et al. (2017). To gain some intuition, if encoder is nearly a constant function, we need to make $K$ and $k$ very large for far away pair of $x$ and $y$, in order for the first inequality in the lemma to hold. While estimating $K$ and $k$ are very interesting, it is far beyond the scope of the present paper.

*Proof.* **Co-Lipschtizness $\implies$ LHS of Equation 18.** Given any $x_1$ and $x_2$, we want to estimate $\mathrm{d}(x_1, x_2)$. We denote their corresponding images: $(y_1 = f(x_1), y_2 = f(x_2))$. Consider $R = \mathrm{d}_1(y_1, y_2)$. By co-Lipschtizness:

$$\mathrm{Diameter}(f^{-1}(B_{\mathrm{d}_1(y_1,y_2)}(y_1))) \leq K \cdot \mathrm{Diameter}(B_{\mathrm{d}_1(y_1,y_2)}(y_1)) + k \tag{21}$$

$$= 2K \cdot \mathrm{d}_1(y_1, y_2) + k \tag{22}$$

By construction, $f^{-1}(B_{\mathrm{d}_1(y_1,y_2)}(y_1))$ includes both $x_1$ and $x_2$. Thus,

$$\mathrm{d}_1(x_1, x_2) \leq \mathrm{Diameter}(f^{-1}(B_{\mathrm{d}_1(y_1,y_2)}(y_1))) \leq 2K \cdot \mathrm{d}_1(f(x_1), f(x_2)) + k \tag{23}$$

**LHS of Equation 18 $\implies$ Co-Lipschtizness.** For any $y$, $R \geq 0$, we want to estimate $\mathrm{Diameter}(f^{-1}(B_R(y)))$. Without loss of generality (otherwise, use a limit argument), assume the existence of $x_1, x_2$ such that $\mathrm{d}(x_1, x_2) = \mathrm{Diameter}(f^{-1}(B_R(y)))$. By the definition of LHS and definition of Diameter,

$$\mathrm{Diameter}(f^{-1}(B_R(y))) \tag{24}$$

$$= \mathrm{d}(x_1, x_2) \tag{25}$$

$$\leq K \mathrm{d}_2(f(x_1), f(x_2)) + k \tag{26}$$

$$\leq K \cdot \mathrm{Diameter}(B_R(y)) + k \tag{27}$$

$$\square$$

In other words, requiring the encoder mapping $\mathrm{Enc} : \mathbf{x} \rightarrow (\mu(\mathbf{x}), \sigma(\mathbf{x}))$ to be co-Lipschitz (co-coarse-Lipschiz) with degrees $(K, k)$ is to ask $\mathrm{Enc}$ to obey the LHS of the quasi-isometry inequality 18.

$$\forall \mathbf{x}_1, \mathbf{x}_2 \in \mathrm{ESupt}(P_{\mathrm{IID}}) \cup \mathrm{ESupt}(P_{\mathrm{OOD}}) : \tag{28}$$

$$\frac{1}{K} \mathrm{d}_1(\mathbf{x}_1, \mathbf{x}_2) - \frac{k}{K} \leq \mathrm{d}_2(\mathrm{Enc}(\mathbf{x}_1), \mathrm{Enc}(\mathbf{x}_2)) \tag{29}$$

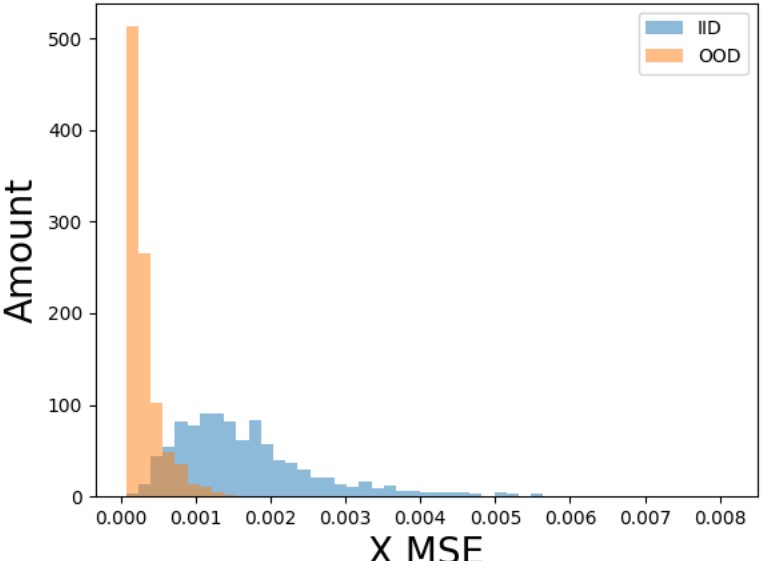

Figure 6: **Small test time reconstruction for IID.**

On the other hand, if $\mathrm{Dec} : \mathbf{z} \to (\mu(\mathbf{z}), \sigma(\mathbf{z}))$ is Lipschiz or coarse-Lipschitz,

$$\forall \mathbf{z}_1, \mathbf{z}_2 \in \mathrm{Enc}(\mathrm{ESupt}(P_{\mathrm{IID}})) \cup \mathrm{Enc}(\mathrm{ESupt}(P_{\mathrm{OOD}})) : \tag{30}$$

$$\mathrm{d}_2(\mathbf{z}_1, \mathbf{z}_2) \le L \mathrm{d}_1(\mathrm{Dec}(\mathbf{z}_1), \mathrm{Dec}(\mathbf{z}_2)) + l \tag{31}$$

$$\mathrm{d}_2(\mathrm{Enc}(\mathbf{x}_1), \mathrm{Enc}(\mathbf{x}_2)) \le L \mathrm{d}_1(\mathrm{Dec}(\mathrm{Enc}(\mathbf{x}_1)), \mathrm{Dec}(\mathrm{Enc}(\mathbf{x}_2))) + l \tag{32}$$

Put together:

$$\forall \mathbf{x}_1, \mathbf{x}_2 \in \mathrm{ESupt}(P_{\mathrm{IID}}) \cup \mathrm{ESupt}(P_{\mathrm{OOD}}) : \tag{33}$$

$$\frac{1}{K} \mathrm{d}_1(\mathbf{x}_1, \mathbf{x}_2) - \frac{k}{K} \tag{34}$$

$$\le \mathrm{d}_2(\mathrm{Enc}(\mathbf{x}_1), \mathrm{Enc}(\mathbf{x}_2)) \tag{35}$$

$$\le L \mathrm{d}_1(\mathrm{Dec}(\mathrm{Enc}(\mathbf{x}_1)), \mathrm{Dec}(\mathrm{Enc}(\mathbf{x}_2))) + l \tag{36}$$

If $\mathrm{d}_1(\mathrm{Dec}(\mathrm{Enc}(\mathbf{x}_1)), \mathrm{Dec}(\mathrm{Enc}(\mathbf{x}_2))) \approx \mathrm{d}_1(\mathbf{x}_1, \mathbf{x}_2)$, which can be verified empirically for all VAEs, we observe that VAEs' unique encoder-decoder structure tries to learn a probabilistic relaxed version of quasi-isometry with different parametric constants $(K, k)$ and $(L, l)$ on both sides of the inequalities. As a by-product of our theory, we reveal VAEs' nearly quasi-geometric learning behaviour.

### B.3 SUPPLEMENTARY MATERIALS FOR SECTION 3.1.3

In this section, we restate and prove the Theorem 3.8 rigorously. While the proof utilizes some apriori estimates, the main idea can be illustrated in the following Figure 7:

**Theorem B.13** (Provable OOD detection bounds, Theorem 3.8 in the main text). *Fix $P_{IID}$, $P_{OOD}$ and $m_{intra} > 0$ and choose $m_{inter} > 2 \cdot m_{intra}$. Assume without loss of generality the corresponding* $\arg\min$ *in Definition 3.4 for $m_{inter}$ exists, denoted as:* $(\epsilon_{IID}^*, \epsilon_{OOD}^*)$.

*Suppose the encoder $q_\phi : \mathbf{x} \longrightarrow (\mu_{\mathbf{z}}(\mathbf{x}), \sigma_{\mathbf{z}}(\mathbf{x}))$ is co-Lipschitz with degrees $(K, k)$, or the decoder $p_\theta : \mathbf{z} \longrightarrow (\mu_{\mathbf{x}}(\mathbf{z}), \sigma_{\mathbf{x}}(\mathbf{z}))$ is $L$ Lipschitz with $L \le K$ [11]. Then for any metric in the input space $\mathrm{d}_X(\cdot, \cdot)$ [12] upon which $m_{inter}$ and $m_{intra}$ margins are defined, with probability $\ge 1 - (\epsilon_{IID}^* + \epsilon_{OOD}^*)$*

---

[11]This condition is evoked when $q_\phi$ fails to be co-Lipschitz with degrees $(K, k)$. $L \le K$ is sensible because VAEs learns to reconstruct $P_{\mathrm{IID}}$.

[12]We mean metric spaces that obey the triangle inequality. This is extremely general, including widely used $l^\infty$ in adversarial robustness, perceptual distance in vision Gatys et al. (2016), etc. Our result also extends to any metric in the latent spaces. We use $l^2$ norm for the latent variable parameters for simplicity.

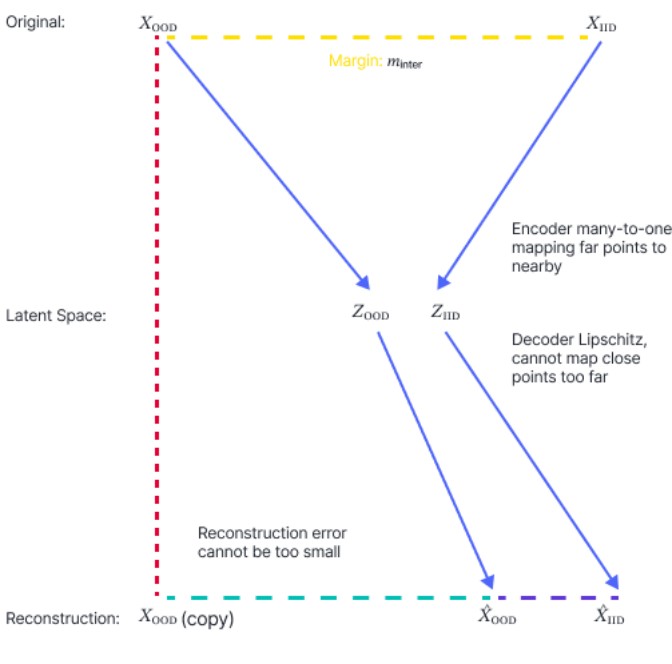

Figure 7: **Proof by geometry.**

*over the joint distribution ($P_{IID}$, $P_{OOD}$), at least one of the following holds:*

$$\|\mu_{\mathbf{z}}(\mathbf{x}_{IID}) - \mu_{\mathbf{z}}(\mathbf{x}_{OOD})\|_2 \geq \frac{m_{inter} - k}{K} \quad and \quad \|\sigma_{\mathbf{z}}(\mathbf{x}_{IID}) - \sigma_{\mathbf{z}}(\mathbf{x}_{OOD})\|_2 \geq \frac{m_{inter} - k}{K} \quad (37)$$

$$\mathrm{d}_X(\mathbf{x}_{OOD}, \widehat{\mathbf{x}}_{OOD}) \geq \frac{2K - L}{2K} m_{inter} - m_{intra} + \frac{kL}{2K} \quad (38)$$

*Proof.* We begin by proving the first inequality when the encoder's latent code $\mu_{\mathbf{z}}$ is co-Lipschitz with degrees $(K, k)$. Recall by definition, for any $\mathbf{y}$:

$$\mathrm{Diameter}((\mu_{\mathbf{z}}, \sigma_{\mathbf{z}})^{-1}(B_R(\mathbf{y}))) \leq K \cdot \mathrm{Diameter}(B_R(\mathbf{y})) + k \quad (39)$$

We denote $\mathbf{z}_{IID} = (\mu_{\mathbf{z}}(\mathbf{x}_{IID}), \sigma_{\mathbf{z}}(\mathbf{x}_{IID}))$ and $\mathbf{z}_{OOD} = (\mu_{\mathbf{z}}(\mathbf{x}_{OOD}), \sigma_{\mathbf{z}}(\mathbf{x}_{OOD}))$. Plugging $R = \|\mathbf{z}_{IID} - \mathbf{z}_{OOD}\|/2$, $\mathbf{y} = \mathbf{z}_{IID}$ to the above inequality, we have:

$$\mathrm{Diameter}((\mu_{\mathbf{z}}, \sigma_{\mathbf{z}})^{-1}(B_{\|\mathbf{z}_{IID} - \mathbf{z}_{OOD}\|/2}(\mathbf{z}_{IID}))) \leq K \cdot \mathrm{Diameter}(B_{\|\mathbf{z}_{IID} - \mathbf{z}_{OOD}\|/2}(\mathbf{z}_{IID})) + k \quad (40)$$

which by the definition of Diameter, simplifies the right hand side (RHS) to:

$$\mathrm{Diameter}((\mu_{\mathbf{z}}, \sigma_{\mathbf{z}})^{-1}(B_{\|\mathbf{z}_{IID} - \mathbf{y}_{OOD}\|/2}(\mathbf{z}_{IID}))) \leq K \cdot \|\mathbf{z}_{IID} - \mathbf{z}_{OOD}\| + k \quad (41)$$

Note that by assumption, $\mathrm{D}_{X|m_{inter}}(P_{IID}, P_{OOD}) \geq m_{inter}$. Thus with probability at least $1 - (\epsilon_{IID}^* + \epsilon_{OOD}^*)$ over the joint distribution ($P_{IID}, P_{OOD}$), $(\mu_{\mathbf{z}}, \sigma_{\mathbf{z}})^{-1}(B_{\|\mathbf{z}_{IID} - \mathbf{z}_{OOD}\|/2}(\mathbf{z}_{IID}))$ contains $\mathbf{x}_{IID}$ and $\mathbf{x}_{OOD}$ that are $m_{inter}$ apart. This translates the above deterministic inequality to the following. With probability at least $1 - (\epsilon_{IID}^* + \epsilon_{OOD}^*)$ over ($P_{IID}, P_{OOD}$), we have:

$$m_{inter} \leq \mathrm{Diameter}((\mu_{\mathbf{z}}, \sigma_{\mathbf{z}})^{-1}(B_{\|\mathbf{z}_{IID} - \mathbf{z}_{OOD}\|/2}(\mathbf{z}_{IID}))) \leq K \cdot \|\mathbf{z}_{IID} - \mathbf{z}_{OOD}\| + k \quad (42)$$

As a result, with probability at least $1 - (\epsilon_{IID}^* + \epsilon_{OOD}^*)$ over ($P_{IID}, P_{OOD}$):

$$\|(\mu_{\mathbf{z}}(\mathbf{x}_{IID}), \sigma_{\mathbf{z}}(\mathbf{x}_{IID})) - (\mu_{\mathbf{z}}(\mathbf{x}_{OOD}), \sigma_{\mathbf{z}}(\mathbf{x}_{OOD}))\|_2 \geq \frac{m_{inter} - k}{K} \quad (43)$$

We remark that the above proof does not utilize the full strength of co-Lipschitzness; we merely use it between $P_{IID}$ and $P_{OOD}$. We also note that the above does not use any particular properties of the $l^2$ norm, and extends to any metric $\mathrm{d}_Z(\cdot, \cdot)$ in the latent spaces.

Next, we prove the second inequality. We will break down the proof into cases. First, we observe that the second inequality is more interesting when the first inequality fails. Otherwise, the first inequality can give an OOD detection score with high probability. We will henceforth use the fact that encoder $q_\phi$ is anti co-Lipschtiz with degree $(K, k)$.

**Case 1 [Encoder is anti co-Lipschtiz within $P_{\textbf{IID}}$ or $P_{\textbf{OOD}}$].** In this case, if encoder remains co-Lipschtiz between $P_{\text{IID}}$ and $P_{\text{OOD}}$, the first inequality is unaffected. And our statement holds trivially.

**Case 2 [Encoder is anti co-Lipschtiz between $P_{\textbf{IID}}$ and $P_{\textbf{OOD}}$].** In this case, we will need the second condition where decoder is assumed to be Lipschitz. By assumption in this case, we also have encoder is anti co-Lipschitz. Thus, we have the following inequalities:

Since the encoder is anti co-Lipsthiz with degrees $(K, k)$, there exist $R > 0$, $\mathbf{z}_{\textbf{IID}}$ and $\mathbf{z}_{\textbf{OOD}}$ such that: for some $\mathbf{x}_{\textbf{IID}} \in (\mu_{\mathbf{z}}, \sigma_{\mathbf{z}})^{-1}(B_R(\mathbf{z}_{\textbf{IID}}))$ and $\mathbf{x}_{\textbf{OOD}} \in (\mu_{\mathbf{z}}, \sigma_{\mathbf{z}})^{-1}(B_R(\mathbf{z}_{\textbf{IID}}))$, we have:

$$d_X(\mathbf{x}_{\textbf{IID}}, \mathbf{x}_{\textbf{OOD}}) > K \cdot \text{Diameter}(B_R(\mathbf{z}_{\textbf{IID}})) + k \tag{44}$$

which implies in particular:

$$d_X(\mathbf{x}_{\textbf{IID}}, \mathbf{x}_{\textbf{OOD}}) > 2K \cdot \|\mathbf{z}_{\textbf{IID}} - \mathbf{z}_{\textbf{OOD}}\|_2 + k \tag{45}$$

Since the decoder is L-Lipschitz, we have: for every $\mathbf{z}_{\textbf{IID}}$ and $\mathbf{z}_{\textbf{OOD}}$, we have:

$$d_X(\widehat{\mathbf{x}}_{\text{IID}}, \widehat{\mathbf{x}}_{\text{OOD}}) \tag{46}$$
$$= d_X((\mu_{\mathbf{x}}, \sigma_{\mathbf{x}})(\mathbf{z}_{\textbf{IID}}), (\mu_{\mathbf{x}}, \sigma_{\mathbf{x}})(\mathbf{z}_{\textbf{OOD}})) \tag{47}$$
$$\leq L\|\mathbf{z}_{\textbf{IID}} - \mathbf{z}_{\textbf{OOD}}\|_2 \tag{48}$$
$$\leq \frac{L}{2K}(d_X(\mathbf{x}_{\textbf{IID}}, \mathbf{x}_{\textbf{OOD}}) - k) \tag{49}$$
$$< \frac{1}{2}d_X(\mathbf{x}_{\textbf{IID}}, \mathbf{x}_{\textbf{OOD}}) \tag{50}$$

We call the above a-prior estimate, and it will be useful for our second apriori estimate. We want to establish another apriori estimate on $d_X(\mathbf{x}_{\text{IID}}, \widehat{\mathbf{x}}_{\text{OOD}})$.

For any metric $d_X$, by the assumption of Definition 3.4, with probability at least $1 - (\epsilon^*_{\text{IID}} + \epsilon^*_{\text{OOD}})$ over $(P_{\text{IID}}, P_{\text{OOD}})$, we can estimate the following:

$$d_X(\mathbf{x}_{\text{IID}}, \widehat{\mathbf{x}}_{\text{OOD}}) \tag{51}$$
$$\leq d_X(\mathbf{x}_{\text{IID}}, \widehat{\mathbf{x}}_{\text{IID}}) + d_X(\widehat{\mathbf{x}}_{\text{IID}}, \widehat{\mathbf{x}}_{\text{OOD}}) \tag{52}$$
$$\leq m_{\text{intra}} + \frac{L}{2K}(d_X(\mathbf{x}_{\textbf{IID}}, \mathbf{x}_{\textbf{OOD}}) - k) \tag{53}$$
$$< \frac{m_{\text{inter}}}{2} + \frac{1}{2}(d_X(\mathbf{x}_{\textbf{IID}}, \mathbf{x}_{\textbf{OOD}})) \tag{54}$$
$$\leq d_X(\mathbf{x}_{\textbf{IID}}, \mathbf{x}_{\textbf{OOD}}) \tag{55}$$

where the first inequality follows from triangle inequality, and the last inequality is where we invoke the essential separation properties, that requires a probabilistic statement.

Now we estimate $d_X(\mathbf{x}_{\text{OOD}}, \widehat{\mathbf{x}}_{\text{OOD}})$ using reverse-triangle inequality repeatedly.

$$d_X(\mathbf{x}_{\text{OOD}}, \widehat{\mathbf{x}}_{\text{OOD}}) \tag{56}$$
$$\geq \left| d_X(\mathbf{x}_{\text{OOD}}, \mathbf{x}_{\text{IID}}) - d_X(\mathbf{x}_{\text{IID}}, \widehat{\mathbf{x}}_{\text{OOD}}) \right| \tag{57}$$
$$= d_X(\mathbf{x}_{\text{OOD}}, \mathbf{x}_{\text{IID}}) - d_X(\mathbf{x}_{\text{IID}}, \widehat{\mathbf{x}}_{\text{OOD}}) \tag{58}$$
$$\geq d_X(\mathbf{x}_{\text{OOD}}, \mathbf{x}_{\text{IID}}) - m_{\text{intra}} - \frac{L}{2K}(d_X(\mathbf{x}_{\textbf{IID}}, \mathbf{x}_{\textbf{OOD}}) - k) \tag{59}$$
$$\geq \frac{2K - L}{2K}m_{\text{inter}} - m_{\text{intra}} + \frac{kL}{2K} \tag{60}$$

where the first inequality follows from reverse triangle inequality, the second equality allows us to remove the absolute sign due to our second aprior estimate (Equation 55), the second inequality follows from our first aprior estimate (Equation 53). This holds with probability at least $1 - (\epsilon^*_{\text{IID}} + \epsilon^*_{\text{OOD}})$ over $(P_{\text{IID}}, P_{\text{OOD}})$, because Equation 55 uses the essential distance or margin. $\qquad\square$

**Discussions and Remarks**

Theorem 3.8 identifies a set of *more relaxed or weaker solution concepts* to OOD detection. Instead of aiming for perfect density estimation, we can try to reduce $(K, k)$ for better **latent code separation**, enlarge $K$ and reduce $L$ for **reconstruction based separation**, or smaller $m_{\text{intra}}$. We investigate and exploit such inevitable trade-offs on $K$ in Sections B.4.2 and B.4.3, which in particular leads to our LPath-2M algorithm (Section 4). Nevertheless, not requiring perfect density doesn't imply our method doesn't benefit from it. Recall $\log p_\theta(\mathbf{x}) \approx \log \left[ \frac{1}{K} \sum_{k=1}^{K} \frac{p_\theta(\mathbf{x}|\mathbf{z}^k) p(\mathbf{z}^k)}{q_\phi(\mathbf{z}^k|\mathbf{x})} \right]$. Better $\log p_\theta(\mathbf{x})$ estimation means it is higher on IID samples and lower on OOD region. This translates to higher $p_\theta(\mathbf{x} \mid \mathbf{z}^k)$ and $p(\mathbf{z}^k)$ for IID, lower on OOD, or both. In other words, we'd expect lower $\|\mathbf{x}_{\text{OOD}} - \widehat{\mathbf{x}}_{\text{OOD}}\|_2$, higher $\|(\mu_{\mathbf{z}}(\mathbf{x}_{\text{IID}}), \sigma_{\mathbf{z}}(\mathbf{x}_{\text{IID}})) - (\mu_{\mathbf{z}}(\mathbf{x}_{\text{OOD}}), \sigma_{\mathbf{z}}(\mathbf{x}_{\text{OOD}}))\|_2$, or both. As a result, our method can benefit from improved density estimation and remain robust when $p_\theta(\mathbf{x})$ estimation is difficult.

Like other Lipschitz conditions in machine learning [13], the co-Lipshitzness of the encoder $q_\phi$ and the Lipshitzness of the decoder $p_\theta$ are theoretical quantities that are difficult to check. Moreover, it is unclear how to enforce them. We propose some heuristics for *encouraging* these conditions in Section B.4.2 and evaluate them empirically in Section 5. The statistical aspects of the geometrically distilled sufficient statistics in Theorem 3.8 are discussed in greater details in Appendix B.5. While some hard-to-evaluate quantities are involved, this may be the first provable result in the unsupervised OOD detection problem. The importance of such theorems stem from the OOD setting. Unlike the IID case, where we can reliably evaluate an algorithm's generalization performance, there is no way control the streaming OOD data. A provable method that comes with theoretical guarantees or limitations is therefore particularly desired.

## B.4 SUPPLEMENTARY MATERIALS FOR SECTION 3.2

### B.4.1 JUSTIFICATION OF THE STATISTICS $\|\mu_{\mathbf{z}}(\mathbf{x}_{\text{OOD}})\|$

Since $\|\mu_{\mathbf{z}}(\mathbf{x}_{\text{IID}}) - \mu_{\mathbf{z}}(\mathbf{x}_{\text{OOD}})\|_2$ involves sampling from $P_{\text{IID}}$, we replace it by $\|\mu_{\mathbf{z}}(\mathbf{x}_{\text{OOD}}) - r_0\|$ in the main paper. In this section, we unfold its relation to Theorem 3.8. We formalize the empirical observation first mentioned Figure

**Assumption B.14** (Concentration of Latent Code Parameters). Let $q_\phi : \mathbf{x} \longrightarrow (\mu_{\mathbf{z}}(\mathbf{x}), \sigma_{\mathbf{z}}(\mathbf{x}))$ denote the encoder latent parameter mapping from the input space. We say, the latent codes concentrate on spherical shells $\mathcal{S}_{\mu(\mathbf{z})}(0, r_0)$ and $\mathcal{S}_{\sigma(\mathbf{z})}(I, r_I)$ centered at $0$ and $I$ with radii $r_0 > 0$ and $r_I > 0$, if for every $\epsilon > 0$ and every $\mathbf{x}_{\text{IID}} \in \text{ESupt}(P_{\text{IID}})$:

$$P_{\text{IID}}(|r_0 - \|\mu_{\mathbf{z}}(\mathbf{x}_{\text{IID}})\|| \geq \epsilon) \leq \frac{C_0(P_{\text{IID}})}{\gamma(\epsilon)} \tag{61}$$

$$P_{\text{IID}}(|r_I - \|\sigma_{\mathbf{z}}(\mathbf{x}_{\text{IID}})\|| \geq \epsilon) \leq \frac{C_I(P_{\text{IID}})}{\gamma(\epsilon)} \tag{62}$$

where $\gamma$ is a strictly monotonic increasing function and $C_0(P_{\text{IID}})$ and $C_I(P_{\text{IID}})$ are constants depending only on the distribution $P_{\text{IID}}$ (e.g. variance of $P_{\text{IID}}$).

We also need the following definition for the below proof:

**Definition B.15** (Projection in metric spaces). $\text{Proj}_Y(x) := \arg\min_{y \in Y} d(y, X)$ denote the projection of $x \in X$ onto $Y$.

For an concrete example, $\text{Proj}_{\mathcal{S}_{\mathbf{z}}}(\mu_{\mathbf{z}}(\mathbf{x}_{\text{OOD}})) := \arg\min_{\mathbf{y} \in \mathcal{S}_{\mathbf{z}}} \|\mathbf{y} - \mu_{\mathbf{z}}(\mathbf{x}_{\text{OOD}})\|$ denote the projection of $\mu_{\mathbf{z}}(\mathbf{x}_{\text{OOD}})$ onto $\mathcal{S}_{\mathbf{z}}$. In other words, $\text{Proj}_{\mathcal{S}_{\mathbf{z}}}(\mu_{\mathbf{z}}(\mathbf{x}_{\text{OOD}}))$ maps $\mu_{\mathbf{z}}(\mathbf{x}_{\text{OOD}})$ to its closest point on $\mathcal{S}_{\mathbf{z}}$. $\arg\min$ is achieved because $\mathcal{S}_{\mathbf{z}}$ is a complete metric space.

**Proposition B.16** (Performance guarantee for $\|\mu_{\mathbf{z}}(\mathbf{x}_{\text{OOD}}) - r_0\|$). *Fix $\epsilon > 0$. Under the conditions of Theorem 3.8, and Assumption B.14, with probability at least $1 - (\epsilon^*_{IID} + \epsilon^*_{OOD}) - \frac{C_0(P_{IID})}{\gamma(\epsilon)}$ for $\mu_{\mathbf{z}}$*

---

[13]For example, Lipschitz gradient condition in the optimization literature, i.e. stochastic gradient descent converges depending on the unknown Lipschitz constant.

and $1 - (\epsilon^*_{IID} + \epsilon^*_{OOD}) - \frac{C_I(P_{IID})}{\gamma(\epsilon)}$ *for* $\sigma_{\mathbf{z}}$, *we have the following inequalities:*

$$|r_0 - \|\mu_{\mathbf{z}}(\mathbf{x}_{OOD})\|| \geq \frac{m_{inter} - k}{K} - \epsilon \tag{63}$$

$$|r_I - \|\sigma_{\mathbf{z}}(\mathbf{x}_{OOD})\|| \geq \frac{m_{inter} - k}{K} - \epsilon \tag{64}$$

*Proof.* First, we use the distance the relation between $\mu_{\mathbf{z}}(\mathbf{x}_{\text{OOD}})$ and $\mu_{\mathbf{z}}(\mathbf{x}_{\text{IID}})$ (Figure 5), and apply the reverse triangle inequality:

$$|r_0 - \|\mu_{\mathbf{z}}(\mathbf{x}_{\text{OOD}})\|| \tag{65}$$

$$= \|\text{Proj}_{\mathcal{S}_{\mu(\mathbf{z})}}(\mu_{\mathbf{z}}(\mathbf{x}_{\text{OOD}})) - \mu_{\mathbf{z}}(\mathbf{x}_{\text{OOD}})\| \tag{66}$$

$$= \|\text{Proj}_{\mathcal{S}_{\mu(\mathbf{z})}}(\mu_{\mathbf{z}}(\mathbf{x}_{\text{OOD}})) - \mu_{\mathbf{z}}(\mathbf{x}_{\text{IID}}) + \mu_{\mathbf{z}}(\mathbf{x}_{\text{IID}}) - \mu_{\mathbf{z}}(\mathbf{x}_{\text{OOD}})\| \tag{67}$$

$$\geq |\|\text{Proj}_{\mathcal{S}_{\mu(\mathbf{z})}}(\mu_{\mathbf{z}}(\mathbf{x}_{\text{OOD}})) - \mu_{\mathbf{z}}(\mathbf{x}_{\text{IID}})\| - \|\mu_{\mathbf{z}}(\mathbf{x}_{\text{IID}}) - \mu_{\mathbf{z}}(\mathbf{x}_{\text{OOD}})\|| \tag{68}$$

$$\tag{69}$$

Next, by Assumption B.14, with probability at least $1 - \frac{C_0(P_{\text{IID}})}{\gamma(\epsilon)}$:

$$|\|\text{Proj}_{\mathcal{S}_{\mu(\mathbf{z})}}(\mu_{\mathbf{z}}(\mathbf{x}_{\text{OOD}})) - \mu_{\mathbf{z}}(\mathbf{x}_{\text{IID}})\| - \|\mu_{\mathbf{z}}(\mathbf{x}_{\text{IID}}) - \mu_{\mathbf{z}}(\mathbf{x}_{\text{OOD}})\|| \tag{70}$$

$$\geq \|\mu_{\mathbf{z}}(\mathbf{x}_{\text{IID}}) - \mu_{\mathbf{z}}(\mathbf{x}_{\text{OOD}})\| - \epsilon \tag{71}$$

Now we can apply Theorem 3.8 to the first term. As a result, with probability at least $(1 - (\epsilon^*_{\text{IID}} + \epsilon^*_{\text{OOD}}))(1 - \frac{C_0(P_{\text{IID}})}{\gamma(\epsilon)}) = 1 - (\epsilon^*_{\text{IID}} + \epsilon^*_{\text{OOD}}) - \frac{C_0(P_{\text{IID}})}{\gamma(\epsilon)} + (\epsilon^*_{\text{IID}} + \epsilon^*_{\text{OOD}})(\frac{C_0(P_{\text{IID}})}{\gamma(\epsilon)})$, we have the following:

$$|r_0 - \|\mu_{\mathbf{z}}(\mathbf{x}_{\text{OOD}})\|| \geq \frac{m_{\text{inter}} - k}{K} - \epsilon \tag{72}$$

The proof for the $\sigma_{\mathbf{z}}(\mathbf{x}_{\text{OOD}})$ is similar and we omit it. $\qquad\square$

**Corollary B.17** (Performance guarantee for $\|\mu_{\mathbf{z}}(\mathbf{x}_{\text{OOD}})\|$). *Under the condition of Proposition B.16, with probability at least* $1 - (\epsilon^*_{IID} + \epsilon^*_{OOD}) - \frac{C_0(P_{IID})}{\gamma(\epsilon)}$ *for* $\mu_{\mathbf{z}}$ *and* $1 - (\epsilon^*_{IID} + \epsilon^*_{OOD}) - \frac{C_I(P_{IID})}{\gamma(\epsilon)}$ *for* $\sigma_{\mathbf{z}}$, *we have the following inequalities:*

$$\|\mu_{\mathbf{z}}(\mathbf{x}_{OOD})\| \geq r_0 + \frac{m_{inter} - k}{K} - \epsilon \quad or \quad \|\mu_{\mathbf{z}}(\mathbf{x}_{OOD})\| \leq r_0 - \frac{m_{inter} - k}{K} + \epsilon \tag{73}$$

$$\|\sigma_{\mathbf{z}}(\mathbf{x}_{OOD})\| \geq r_0 + \frac{m_{inter} - k}{K} - \epsilon \quad or \quad \|\sigma_{\mathbf{z}}(\mathbf{x}_{OOD})\| \leq r_0 - \frac{m_{inter} - k}{K} + \epsilon \tag{74}$$

These are to be compared with Assumption B.14 that characterize the corresponding norms for $P_{\text{IID}}$. As a result, our approximation statistics also enjoy provable properties.

### B.4.2 NOT ALL VAEs ARE BROKEN THE SAME: ENCODER, DECODER AND LATENT DIMENSION

Theorem 3.8 also has quantitative implications on algorithmic design: we may choose VAEs training hyperparameters to empirically optimize OOD performances by searching though $K, k, L$. While it is unclear how to make $q_\phi$ more co-Lipschitz, we can avoid conditions that break it. By the same argument, we want to avoid cases that make $p_\theta$ less Lipschitz continuous. In this section, we discuss heuristics inspired by Theorem 3.8 for training VAEs in the setting of OOD detection.

**The higher the latent dimension, the better encoder can discriminate against OOD.** Theorem 3.8 prefers $q_\phi$ to be region-wise one-to-one. Formally, Equation 37 in Theorem 3.8 suggests we can make the latent code between IID and OOD cases more separable if both $K$ and $k$ are smaller. In other words, when the encoder has small co-Lipschitz degrees. While it is unclear how to make encoder region-wise one-to-one, we identify a condition on the latent dimension (of $\mathbf{z}$) that can make $q_\phi$ fail to be region-wise one-to-one. This condition is on the latent code dimension $m$, which can make $K$ or $k$ arbitrarily large and we would like to avoid it.

**Lemma B.18** (Continuous maps and region-wise one-to-one). *Let* $n < m$. *There exists continuous* $f : \mathbf{z} \in \mathbb{R}^m \longrightarrow \mathbb{R}^n$ *such that it is not region-wise one-to-one with any degrees.*

*Proof.* By the proof of Theorem 1 in Landweber et al. (2016), for any $M > 0$, there is a Lipschitz continuous map $f$ such that $\mathrm{Diameter}(f^{-1}(y)) > M$, for some $y \in \mathbb{R}^n$. In particular, $\mathrm{Diameter}(f^{-1}(B_R(y))) > M$ since the above is a subset of this. Since $\mathrm{Diameter}(f^{-1}(B_R(y)))$ can be arbitrarily large, by choosing $R = 1$, there will be no $(K, k)$ pair in Definition 3.6 that can bound $\mathrm{Diameter}(f^{-1}(B_R(y)))$, proving our claim. □

Setting $f = q_\phi$, Lemma B.18 implies we can no longer confidently rely on Theorem 3.8 to detect OOD, as long as the latent dimension ($m$) is smaller than input ambient dimension ($n$) (e.g. 784 for MNIST). While such pathological cases may not happen in practice, making latent dimension bigger is sensible for OOD detection: as we increase VAEs' latent dimension, $q_\phi$ can find more room so that $\mathbf{x}_{\text{IID}}$ does not mix up with $\mathbf{x}_{\text{OOD}}$ much. More precisely, the "large fiber lemma" and its associated results from Landweber et al. (2016) implies $\mathrm{Diameter}(f^{-1}(\mathbf{y}))$ can be arbitrarily large, whenever target space dimension is smaller than the input's. Letting $f(\mathbf{x}) = \mathbf{y} = (\mu_{\mathbf{z}}(\mathbf{x}), \sigma_{\mathbf{z}}(\mathbf{x}))$ in $q_\phi(\mathbf{z}|\mu_{\mathbf{z}}(\mathbf{x}), \sigma_{\mathbf{z}}(\mathbf{x}))$, since $f^{-1}(\mathbf{y})$ is big in diameter, $f(\mathbf{x}_{\text{IID}})$ and $f(\mathbf{x}_{\text{OOD}})$ can be mapped to nearly the same $\mathbf{y} = (\mu_{\mathbf{z}}(\mathbf{x}), \sigma_{\mathbf{z}}(\mathbf{x}))$, even if $\mathbf{x}_{\text{IID}}$ and $\mathbf{x}_{\text{OOD}}$ are farther away. While setting higher latent dimension is not a sufficient condition for $q_\phi$ to be region-wise one-to-one, not meeting it will make $q_\phi$ susceptible to region-wise one-to-one pathological cases. This mathematical intuition does suggest us to try training VAEs with very high latent dimension for OOD detection, even at the cost of over-fitting, etc.

**The lower the latent dimension, the better decoder screens for OOD.** Theorem 3.8 also prefers $p_\theta$ to have a small Lipschitz constant, i.e. more Lipschitz continuous. More precisely, since VAEs' learning objective is to reconstruct, i.e. learning an approximate identity: $\widehat{\mathbf{x}} \approx \mathbf{x}$, it is sensible to expect $K$ and $L$ to be at the same order of magnitudes. Thus, the dominating term on the right hand side of Equation 38 in Theorem 3.8 is the first term: $\frac{2K - L}{2K} m_{\text{inter}}$. To make this term bigger through VAEs function analytic properties, we can make $L$ smaller and make $K$ bigger. In the prior paragraph, we discussed why encoder prefers smaller $K$ to be better at screening OOD data. This poses a conflicting requirement on $K$. Since $L$ is also at our disposal, our heuristics in this section focuses on $L$.

In the spirit of Definition 3.5, we measure $L$, how continuous $p_\theta(\mathbf{z})$ is, by bounding its Jacobian:

**Lemma B.19** (Jacobian matrix estimates). *Let $f : \mathbf{z} \in \mathbb{R}^m \longrightarrow \mathbb{R}^n$ be any differentiable function. Assume each entry of $\mathrm{J}_{\mathbf{z}} f(\mathbf{z})$ is bounded by some constant $C$. We have $f$ is Lipschitz with Lipschitiz constant $L$:*

$$L = \sup_{\mathbf{z}} \|\mathrm{J}_{\mathbf{z}} f\|_2 := \sup_{\mathbf{z}} \sup_{\mathbf{u} \neq 0} \frac{\|\mathrm{J}_{\mathbf{z}} f \mathbf{u}\|_2}{\|\mathbf{u}\|_2} \leq C\sqrt{m}\sqrt{n} \tag{75}$$

*Proof.* First, if $\|\mathrm{J}_{\mathbf{z}} f\|_2$ is bounded, then $f$ is Lipschitz by the mean value theorem. It suffices to prove Jacobian is bounded. Next, $\|\mathrm{J}_{\mathbf{z}} f\|_2 \leq \sqrt{mn}\|\mathrm{J}_{\mathbf{z}} f\|_{\text{Max}} \leq \sqrt{mn}C$ by the matrix norm equivalence. □

Lemma B.19 suggests one way to globally control $p_\theta$'s modulus of continuity: by making the latent dimension $m$ unusually small (we cannot choose the input dimension.). This will break $p_\theta$' ability to reconstruct $\mathbf{x}_{\text{OOD}}$ well whenever $\mathbf{z}_{\text{OOD}}$ is mapped to near any $\mathbf{z}_{\text{IID}}$. In other words, $\widehat{\mathbf{x}}_{\text{OOD}} = p_\theta(\mathbf{z}_{\text{OOD}}) \approx \widehat{\mathbf{x}}_{\text{IID}}$. In this way, we mix $\widehat{\mathbf{x}}_{\text{IID}}$ and $\widehat{\mathbf{x}}_{\text{OOD}}$ together, leading to large reconstruction errors.

This happens when $\mathbf{z}_{\text{IID}}$ and $\mathbf{z}_{\text{OOD}}$ are mixed together, and $p_\theta$ is Lipschitz continuous, which leads us to rethink OOD's representation learning objective. What makes $u(\mathbf{x})$ an discriminative scoring function for OOD detection? In the OOD detection sense, we want a DGM to learn tailored features to reconstruct IID data well only, while such specialized representations will fail to recover OOD data. *These OOD detection requirements drastically differ from that of conventional supervised and unsupervised learning, that aims to learn universal features (Devlin et al., 2018; He et al., 2016).* While ML research aims for general AI and universal representations, VAE OOD detection seems to ask for the opposite.

Section B.4.2 gives conflicting requirements on the latent dimension $m$ (also see our discussion of it in terms of unclear signs of $K$ (The discussion paragraph after Theorem 3.8)). Making $K$ bigger suggests setting larger $m$, while making $L$ smaller implies setting smaller $m$.

We further discuss how to take advantage of this paradox in Appendix B.4.3, leading us to pair two broken VAEs. To sum our heuristics for OOD detection, we try to encourage bigger $K$ for encoder and smaller $L$ for decoder.

### B.4.3  BROKEN VAEs PAIRING: IT TAKES TWO TO TRANSCEND

**One VAE faces a trade-off in latent dimension: $q_\phi$ wants it to be big while $p_\theta$ wants it small.** Section B.4.2 leaves us with a paradox: enlarging latent dimension $m$ is necessary for $q_\phi$'s region-wise one-to-one, but can allow $p_\theta$ to be less continuous. It does not seem we can leverage this observation in a *single* VAE.

**Two VAEs face no such trade-offs.** We propose to train two VAEs, take the latent dimensionally constrained (small $m$) $p_\theta$'s $u(\mathbf{x})$, get the overparamterized (big $m$) $q_\phi$'s $v(\mathbf{x})$ and $w(\mathbf{x})$, and combine them as the joint statistics for OOD detection. In this way, we avoid the dimensional trade-off in any single VAE. In the very hard cases where a DGM is trained on CIFAR 10 as in-distribution, and CIFAR 100, VFlip and HFlip as OOD, we advanced SOTA empirical results significantly. This is surprising given both VAEs are likely broken with poorly estimated likelihoods. The over-parameterized VAE is likely broken, because it may over-fit more easily (generalization error). The overly constrained one is probably also broken, since it has trouble reconstructing many training data (approximation error). However, together they achieved better performance, even better than much bigger model architectures specifically designed to model image data better. See Table 1.

### B.5  FROM LIKELIHOOD AND SUFFICIENCY PRINCIPLES TO LIKELIHOOD PATH PRINCIPLE

In this section, we show Section 3.1's geometric argument is related to the well-known likelihood and sufficiency principles, applied to encoder and decoder conditional likelihoods. This further solidifies the likelihood path principle in Section 2.

**Screening $\mathbf{x_{OOD}}$ using $\log p_\theta(\mathbf{x_{OOD}})$ alone does not perform explicit statistical inferences.** In the fully unsupervised cases, i.e. Morningstar et al. (2021); Havtorn et al. (2021); Xiao et al. (2020), most OOD detection methods use $\log p_\theta(\mathbf{x})$ or its close cousins to screen, instead of performing explicit hypothesis testing. This is probably because $p_\theta(\mathbf{x})$ is parameterized by neural nets, having no closed form. In particular, $p_\theta(\mathbf{x})$ doesn't have an *instance dependent* parameter to be tested against in test time. Thus, it is less clear what inferences are performed to test the IID v.s. OOD hypothesis.

**Latent variable models can perform instance dependent statistical inferences.** On the other hand, latent variable DGMs such as Gaussian VAE, perform explicit statistical inferences on latent parameters $\mu_{\mathbf{z}}(\mathbf{x}), \sigma_{\mathbf{z}}(\mathbf{x})$ in the encoder $q_\phi(\mathbf{z}|\mu_{\mathbf{z}}(\mathbf{x}), \sigma_{\mathbf{z}}(\mathbf{x}))$. Then after observing $\mathbf{z}_k \sim q_\phi(\mathbf{z}|\mu_{\mathbf{z}}(\mathbf{x}), \sigma_{\mathbf{z}}(\mathbf{x}))$, $\mu_{\mathbf{x}}(\mathbf{z}_k), \sigma_{\mathbf{x}}(\mathbf{z}_k)$ are inferred by the decoder $p_\theta(\mathbf{x}|\mu_{\mathbf{x}}(\mathbf{z}_k), \sigma_{\mathbf{x}}(\mathbf{z}_k))$ in the visible space. In other words, $\mu_{\mathbf{z}}(\mathbf{x}), \sigma_{\mathbf{z}}(\mathbf{x})$ and $\mu_{\mathbf{x}}(\mathbf{z}_k), \sigma_{\mathbf{x}}(\mathbf{z}_k)$ can be interpreted as a hypothesis proposed by VAEs to explain the observations $\mathbf{x}$ and $\mathbf{z}_k$. Standard decision based on $\log p_\theta(\mathbf{x})$ alone, without considering the conditional likelihood path, can lead to information loss (Section 2).

**VAEs' likelihood paths are sufficient for OOD detection, as per likelihood and sufficiency principles.** Applying Equations 37 and 38 can be interpreted as following the *likelihood principle* in both the latent and visible spaces. In the inference about model parameters, after $\mathbf{x}$ or $\mathbf{z}_k$ is observed, all relevant information is contained in the conditional likelihood function. Implicitly, there lies the *sufficiency principle*: for two different observations $\mathbf{x}_1$ and $\mathbf{x}_2$ ($\mathbf{z}_1$ and $\mathbf{z}_1$, respectively) having the same values $T(\mathbf{x}_1) = T(\mathbf{x}_2)$ ($T(\mathbf{z}_1) = T(\mathbf{z}_2)$, respectively) of a statistics $T$ sufficient for some model family $p(\cdot|\xi)$, the inferences about $\xi$ based on $\mathbf{x}_1$ and $\mathbf{x}_2$ should be the same. For Gaussian VAEs, a pair of *minimal* sufficient statistics $T$ for $\xi$ is $(\mu_{\mathbf{z}}(\mathbf{x}), \sigma_{\mathbf{z}}(\mathbf{x}))$ for encoder, and $(\mu_{\mathbf{x}}(\mathbf{z}), \sigma_{\mathbf{x}}(\mathbf{z}))$ for decoder respectively. In other words, in the likelihood information theoretic sense, all other information such as neural net intermediate activation is irrelevant for screening $\mathbf{x_{OOD}}$ and $(\mu_{\mathbf{z}}(\mathbf{x}), \sigma_{\mathbf{z}}(\mathbf{x})), \mu_{\mathbf{x}}(\mathbf{z}), \sigma_{\mathbf{x}}(\mathbf{z})$ are sufficiently informative. Therefore, the geometric arguments in Section 3.1 in fact are also grounded in statistical inferences.

Recall the likelihood path principle proposed in Section 1 and Section 2. Our geometric and statistical arguments reveal particularly informative neural activation paths: the minimal sufficient statistics of $p_\theta$ and $q_\phi$. In this case, the likelihood path principle reduces down to likelihood and sufficiency principles for the encoder and decoder likelihoods, because how VAEs estimate $\log p_\theta(\mathbf{x})$ (See Equation 1).

**Framing OOD detection as statistical hypothesis testing.** A rigorous and obvious way of using these inferred parameters is the *likelihood ratio test*. We begin our discussion with the decoder $p_\theta(\mathbf{x}|\mu_{\mathbf{z}_k}(\mathbf{x}), \sigma_{\mathbf{z}_k}(\mathbf{x}))$'s parameter inferences, where $\mathbf{z}_k \sim q_\phi(\mathbf{z}|\mu_{\mathbf{z}}(\mathbf{x}), \sigma_{\mathbf{z}}(\mathbf{x}))$ from the encoder. Since $\mathbf{z}_k \sim q_\phi(\mathbf{z}|\mu_{\mathbf{z}}(\mathbf{x}), \sigma_{\mathbf{z}}(\mathbf{x}))$ is indexed by $\mathbf{x}$, we consider the following average likelihood ratio:

$$\lambda_{\mathrm{LR}}^{\mathbf{x}}(\mathbf{x}) = \log \frac{\mathbb{E}_{\mathbf{z}_k \sim q_\phi(\mathbf{z}|\mu_{\mathbf{z}}(\mathbf{x}), \sigma_{\mathbf{z}}(\mathbf{x}))} p_\theta(\mathbf{x} \mid \mu_{\mathbf{x}}(\mathbf{z}_k), \sigma_{\mathbf{x}}(\mathbf{z}_k))}{\sup_{\mathbf{x}_{\mathrm{IID}}} \mathbb{E}_{\mathbf{z}_l \sim q_\phi(\mathbf{z}|\mu_{\mathbf{z}}(\mathbf{x}_{\mathrm{IID}}), \sigma_{\mathbf{z}}(\mathbf{x}_{\mathrm{IID}}))} p_\theta(\mathbf{x} \mid \mu_{\mathbf{x}}(\mathbf{z}_l), \sigma_{\mathbf{x}}(\mathbf{z}_l))} \tag{76}$$

This tests the goodness of fit of two competing statistical models, the null hypothesis proposed by VAEs: $(\mu_{\mathbf{z}}(\mathbf{x}), \sigma_{\mathbf{z}}(\mathbf{x}))$, v.s. the alternative hypotheses which are the set of all decoder latent code indexed by $\mathbf{x}_{\mathrm{IID}}$, at the observed evidence $\mathbf{x}$. We compare the average decoder density over $\mathbf{z}_k \sim q_\phi(\mathbf{z}|\mu_{\mathbf{z}}(\mathbf{x}), \sigma_{\mathbf{z}}(\mathbf{x}))$ to those indexed by $\mathbf{x}_{\mathrm{IID}}$. If $\mathbf{x}$ comes from the same distribution as $\mathbf{x}_{\mathrm{IID}}$, the two average likelihoods should differ no more than the sampling error. Similarly, we have the following for the latent space:

$$\lambda_{\mathrm{LR}}^{\mathbf{z}}(\mathbf{x}) = \log \frac{\mathbb{E}_{\mathbf{z}_k \sim q_\phi(\mathbf{z}|\mu_{\mathbf{z}}(\mathbf{x}), \sigma_{\mathbf{z}}(\mathbf{x}))} p(\mathbf{z}_k \mid \mu_{\mathbf{z}}(\mathbf{x}), \sigma_{\mathbf{z}}(\mathbf{x}))}{\sup_{\mathbf{x}_{\mathrm{IID}}} \mathbb{E}_{\mathbf{z}_k \sim q_\phi(\mathbf{z}|\mu_{\mathbf{z}}(\mathbf{x}), \sigma_{\mathbf{z}}(\mathbf{x}))} p(\mathbf{z}_k \mid \mu_{\mathbf{z}}(\mathbf{x}_{\mathrm{IID}}), \sigma_{\mathbf{z}}(\mathbf{x}_{\mathrm{IID}}))} \tag{77}$$

As observed in Nalisnick et al., VAEs can assign higher likelihood to OOD data. This can affect the efficacy of Equations 76 and 77. Similar to Morningstar et al. (2021)'s OOD detection approach, the likelihood having wrong orders problem (assigning higher likelihood to OOD samples) can be partially addressed by fitting another classical algorithm on top. We follow the same approach by considering the distribution of $(\lambda_{\mathrm{LR}}^{\mathbf{x}}(\mathbf{x}), \lambda_{\mathrm{LR}}^{\mathbf{z}}(\mathbf{x}))$. In other words, to deal with typicality, which can affect the order of the conditional likelihood ratios, we regard $(\lambda_{\mathrm{LR}}^{\mathbf{x}}(\mathbf{x}), \lambda_{\mathrm{LR}}^{\mathbf{z}}(\mathbf{x}))$ as random variables and use their distributions to discriminate against OOD samples. As a result, ratios that are too small or too big would be considered as OOD.

From a minimal sufficient statistics perspective, instead of $(\lambda_{\mathrm{LR}}^{\mathbf{x}}(\mathbf{x}), \lambda_{\mathrm{LR}}^{\mathbf{z}}(\mathbf{x}))$: we can consider $(\mu_{\mathbf{z}}(\mathbf{x}), \sigma_{\mathbf{z}}(\mathbf{x}), \mu_{\mathbf{x}}(\mathbf{z}), \sigma_{\mathbf{x}}(\mathbf{z}))$ which may enjoy some numerical/arithmetic cancellation advantages, as we shall explain below.

**Relation to Theorem 3.8.** Encoder's Equation 37 corresponds to the $\mathbf{z}$'s Equation 77's numerator, and decoder's Equation 38 corresponds to $\mathbf{x}$'s Equation 76's numerator. In typical VAE learning, decoder's variance is fixed Dai et al., so it cannot be used as an inferential parameter. This reduces the minimal sufficient statistics for encoder and decoder pair:

$$(\mu_{\mathbf{z}}(\mathbf{x}), \sigma_{\mathbf{z}}(\mathbf{x}), \mu_{\mathbf{x}}(\mathbf{z}), \sigma_{\mathbf{x}}(\mathbf{z})) \longrightarrow (\mu_{\mathbf{z}}(\mathbf{x}), \sigma_{\mathbf{z}}(\mathbf{x}), \mu_{\mathbf{x}}(\mathbf{z})) \tag{78}$$

As a result, the decoder variance won't be used. We begin discussing the decoder's remaining inferential parameters. Since we fit a second stage classical statistical algorithm, the magnitude of $\mu_{\mathbf{z}}(\mathbf{x})$ can cause comparison issues. For this reason, it is natural to use the normalized version and its norm instead:

$$\|\mu_{\mathbf{x}}(\mathbf{z}) - \mathbf{x}\|_2 \tag{79}$$

which, up to some constant factor adjustment, is the $\log p_\theta(\mathbf{x}|\mathbf{z})$ in Equation 76.

We can apply the same reasoning to the encoder when $\mathbf{z}_k$ is chosen to be 0, as a one point approximation to $\mathbf{z}_k$ sampling procedure. One justification is that the encoder to regularized to be close to $\mathcal{N}(0, I)$. This is not perfect, but will expedite computation in test time. More importantly, it corresponds to our geometric analysis in Theorem 3.8 nicely. Recall:

$$\inf_{\mathbf{x}_{\mathrm{IID}}, \mathbf{x}_{\mathrm{OOD}}} \|\mu_{\mathbf{z}}(\mathbf{x}_{\mathrm{IID}}) - \mu_{\mathbf{z}}(\mathbf{x}_{\mathrm{OOD}})\|_2 \tag{80}$$

In test time, we won't observe all OOD samples in a full batch. For a single sample, we can approximate the above by the following one point approximation by taking out the $\inf$ over OOD:

$$\inf_{\mathbf{x}_{\mathrm{IID}}, \mathbf{x}_{\mathrm{OOD}}} \|\mu_{\mathbf{z}}(\mathbf{x}_{\mathrm{IID}}) - \mu_{\mathbf{z}}(\mathbf{x}_{\mathrm{OOD}})\|_2 \approx \inf_{\mathbf{x}_{\mathrm{IID}}} \|\mu_{\mathbf{z}}(\mathbf{x}_{\mathrm{IID}}) - \mu_{\mathbf{z}}(\mathbf{x}_{\mathrm{OOD}})\|_2 \tag{81}$$

By the observed concentration phenomenon discussed in Section 3.2 and Figure 5,

$$\inf_{\mathbf{x}_{\mathrm{IID}}} \|\mu_{\mathbf{z}}(\mathbf{x}_{\mathrm{IID}}) - \mu_{\mathbf{z}}(\mathbf{x}_{\mathrm{OOD}})\|_2 \approx |\|\mu_{\mathbf{z}}(\mathbf{x})\|_2 - r_0| \approx \|\mu_{\mathbf{z}}(\mathbf{x})\|_2 \tag{82}$$

where the last approximation is because when screening OOD samples in test time, for all $\mathbf{x}$, be it IID or OOD, $r_0$ is a constant. We can further drop it before feeding this statistics to the second stage statistical algorithm such as COPOD Li et al. (2020). The reasoning for $\sigma_{\mathbf{z}}(\mathbf{x})$ is identical and we omit it here. This relates our remaining test statistics to Equation 77.

### B.5.1 Training Objective Modification for Stronger Concentration

To encourage stronger concentration empirically observed in Section 3.2, we propose the following modifications to standard VAEs' loss functions:

We replace the initial KL divergence by:

$$\mathcal{D}^{\text{typical}}[Q_\phi(\mathbf{z} \mid \mu_{\mathbf{z}}(\mathbf{x}), \sigma(\mathbf{x}))\|P(\mathbf{z})] \tag{83}$$

$$=\mathcal{D}^{\text{typical}}[\mathcal{N}(\mu_{\mathbf{z}}(\mathbf{x}), \sigma_{\mathbf{z}}(\mathbf{x}))\|\mathcal{N}(0, I)] \tag{84}$$

$$=\frac{1}{2}\left(\text{tr}(\sigma_{\mathbf{z}}(\mathbf{x})) + |(\mu_{\mathbf{z}}(\mathbf{x}))^\top(\mu_{\mathbf{z}}(\mathbf{x})) - m| - m - \log\det(\sigma_{\mathbf{z}}(\mathbf{x}))\right) \tag{85}$$

where $m$ is the latent dimension. This will encourage the latent code $\mu(x)$ to concentrate on the spherical shell with radius $\sqrt{m}$. This is chosen due to the well known concentration of Gaussian probability measures.

In training, we also use Maximum Mean Discrepancy (MMD) Gretton et al. (2012) as a discriminator since we are not dealing with complex distribution but Gaussian. The MMD is computed with Gaussian kernel. This extra modification is because the above magnitude regularization does not take distribution in to account.

The final objective:

$$\mathbb{E}_{\mathbf{x}\sim P_{\text{IID}}}\mathbb{E}_{\mathbf{z}\sim Q_\phi}\mathbb{E}_{\mathbf{n}\sim\mathcal{N}}[\log P_\theta(\mathbf{x} \mid \mathbf{z})] - \mathcal{D}^{\text{typical}}[Q_\phi(\mathbf{z} \mid \mu_{\mathbf{z}}(\mathbf{x}), \sigma(\mathbf{x}))\|P(\mathbf{z})] - \text{MMD}(\mathbf{n}, \mu_{\mathbf{z}}(\mathbf{x})) \tag{86}$$

The idea is that for $P_{\text{IID}}$, we encourage the latent codes to concentrate around the prior's *typical sets*. That way, $P_{\text{OOD}}$ may deviate further from $P_{\text{IID}}$ in a controllable manner. In experiments, we tried the combinations of the metric regularizer, $\mathcal{D}^{\text{typical}}$, and the distribution regularizer, MMD. This leads to two other objectives:

$$\mathbb{E}_{\mathbf{x}\sim P_{\text{IID}}}\mathbb{E}_{\mathbf{z}\sim Q_\phi}[\log P_\theta(\mathbf{x} \mid \mathbf{z})] - \mathcal{D}^{\text{typical}}[Q_\phi(\mathbf{z} \mid \mu_{\mathbf{z}}(\mathbf{x}), \sigma(\mathbf{x}))\|P(\mathbf{z})] \tag{87}$$

$$\mathbb{E}_{\mathbf{x}\sim P_{\text{IID}}}\mathbb{E}_{\mathbf{z}\sim Q_\phi}\mathbb{E}_{\mathbf{n}\sim\mathcal{N}}[\log P_\theta(\mathbf{x} \mid \mathbf{z})] - \mathcal{D}[Q_\phi(\mathbf{z} \mid \mu_{\mathbf{z}}(\mathbf{x}), \sigma(\mathbf{x}))\|P(\mathbf{z})] - \text{MMD}(\mathbf{n}, \mu_{\mathbf{z}}(\mathbf{x})) \tag{88}$$

where $\mathcal{D}$ is the standard KL divergence. In Section C.3, we also describe more experimental details. In short, we found the differences insignificant among the different variations. The minimal sufficient statistics are fairly robust for AUC.

## C Experimental Details

### C.1 Feature Processing To Boost COPOD Performances

Like most statistical algorithms, COPOD/MD is not scale invariant, and may prefer more dependency structures closer to the linear ones. When we plot the distributions of $u(\mathbf{x})$ and $v(\mathbf{x})$, we find that they exhibit extreme skewness. To make COPOD's statistical estimation easier, we process them by quantile transform. That is, for IID data, we map the the tuple of statistics' marginal distributions to $\mathcal{N}(0, 1)$. To ease the low dimensional empirical copula, we also de-correlate the joint distribution of $(u(\mathbf{x}), v(\mathbf{x})), w(\mathbf{x}))$. We do so using Kessy et al. (2018)'s de-correlation method, similar to Morningstar et al. (2021).

### C.2 Width and Height of a Vector Instead of Its $l^2$ Norm To Extract Complementary Information

In our visual inspection, we find that the distribution of the scalar components of $(u(\mathbf{x}), v(\mathbf{x}), w(\mathbf{x}))$ can be rather uneven. For example, the visible space reconstruction $\mathbf{x} - \widehat{\mathbf{x}}$ error can be mostly low for many pixels, but very high at certain locations. These information can be washed away by the $l^2$ norm. Instead, we propose to track both $l^p$ norm and $l^q$ norm for small $p$ and large $q$.

**For small $p$, $l^p$ measures the width of a vector, while $l^q$ measures the height of a vector for big $q$.** To get a sense of how they capture complementary information, we can borrow intuition from

$l^p \approx l^0$, for small $p$ and $l^q \approx l^\infty$, for large $q$. $\|\mathbf{x}\|_0$ counts the number of nonzero entries, while $\|\mathbf{x}\|_\infty$ measures the height of $\mathbf{x}$. For $\mathbf{x}$ with continuous values, however, $l^0$ norm is not useful because it always returns the dimension of $\mathbf{x}$, while $l^\infty$ norm just measures the maximum component.

**Extreme measures help screen extreme data.** We therefore use $l^p$ norm and $l^q$ norm as a continuous relaxation to capture this idea: $l^p$ norm will "count" the number of components in $\mathbf{x}$ that are unusually small, and $l^q$ norm "measures" the average height of the few biggest components. These can be more discriminitive against OOD than $l^2$ norm alone, due to the extreme (proxy for OOD) conditions they measure. We observe some minor improvements, detailed in Table 2's ablation study.

| IID: CIFAR10 | | OOD | | |
|---|---|---|---|---|
| OOD Dataset | SVHN | CIFAR100 | Hflip | Vflip |
| $l^2$ norm | 0.96 | 0.60 | **0.53** | **0.61** |
| $(l^p, l^q)$ | **0.99** | **0.62** | **0.53** | **0.61** |

Table 2: Comparing the AUC of $l^2$ norm versus our $(l^p, l^q)$ measures.

### C.3 VAE ARCHITECTURE AND TRAINING

For the architecture and the training of our VAEs, we followed Xiao et al. (2020). In addition, we have trained VAEs of varying latent dimensions, {1, 2, 5, 10, 100, 1000, 2000, 3096, 5000, 10000}, and instead of training for 200 epochs and taking the resulting model checkpoint, we took the checkpoint that had the best validation loss. For LPath-1M, we conducted experiments on VAEs with all latent dimensions and for LPath-2M, we paired one high-dimensional VAE from the group {3096, 5000, 10000} and one low-dimensional VAE from the group {1, 2, 5}.

In addition to Gaussian VAEs as mentioned in Section B.5, we also empirically experimented with a categorical decoder, in the sense the decoder output is between the discrete pixel ranges, as in Xiao et al. (2020). Strictly speaking, this no longer satisfies the Gaussian distribution anymore, which may in turn violate our sufficient statistics perspective. However, we still experimented with it to test whether LPath principles can be interpreted as a heuristic to inspire methods that approximate sufficient statistics that can work reasonably well, and we observed that categorical decoders work similarly with Guassian decoders.

In addition, we also experimented with VAEs with slightly varied training objectives as detailed in Appendix B.5.1 where we added though we did not observe a significant difference in the final AUROC. In Table 1, we report the best test AUROC in our experiments following the convention in prior works.

## D ABLATION STUDIES

### D.1 COPOD ON FOUR CASES

To verify that the dataset can be divided into four cases as depicted in Figure 1, we separate the dataset into four cases and use our methods on each case. We use the modes of the IID and OOD distributions on mse reconstruction ($u(\mathbf{x})$) and the norm of the latent code ($v(\mathbf{x})$) to decide where the two distributions are considered to overlap, see Figure 8 for a visualization.

The four cases correspond to:

Case 1: $\mathbf{z}_{\text{overlap}} + \mathbf{x}_{\text{overlap}}$
Case 2: $\mathbf{z}_{\text{separable}} + \mathbf{x}_{\text{overlap}}$
Case 3: $\mathbf{z}_{\text{overlap}} + \mathbf{x}_{\text{separable}}$
Case 4: $\mathbf{z}_{\text{separable}} + \mathbf{x}_{\text{separable}}$

Results are reported in Table 3. We can see that the order of the performances respect their conjectured level of difficulty. Our method performs considerably better than other statistics, primarily on Case 1. If we make the overlapping region smaller, for example, by using more extreme quantiles, Case 1 will have fewer samples and the OOD detection would become more difficult.

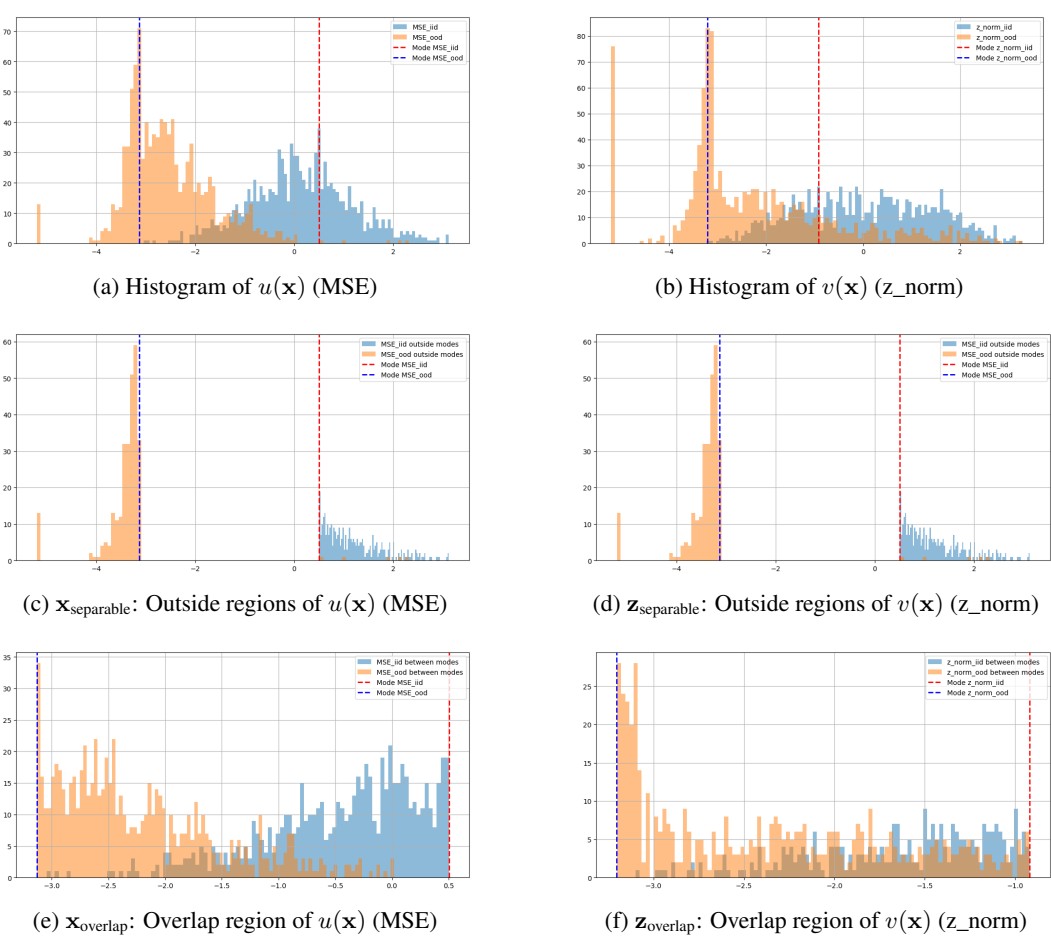

(a) Histogram of $u(\mathbf{x})$ (MSE)

(b) Histogram of $v(\mathbf{x})$ (z_norm)

(c) $\mathbf{x}_{\text{separable}}$: Outside regions of $u(\mathbf{x})$ (MSE)

(d) $\mathbf{z}_{\text{separable}}$: Outside regions of $v(\mathbf{x})$ (z_norm)

(e) $\mathbf{x}_{\text{overlap}}$: Overlap region of $u(\mathbf{x})$ (MSE)

(f) $\mathbf{z}_{\text{overlap}}$: Overlap region of $v(\mathbf{x})$ (z_norm)

Figure 8: How overlap and outside regions are defined in Appendix D.1
.

|  | Case 1 | Case 2 | Case 3 | Case 4 |
|---|---|---|---|---|
| $v(\mathbf{x})$ | 0.75 | 0.74 | 0.93 | 0.96 |
| $u(\mathbf{x})$ | 0.93 | 0.98 | 1.00 | 0.98 |
| ELBO | 0.83 | 0.87 | 1.00 | 0.96 |
| Ours | 0.99 | 0.99 | 1.00 | 0.99 |

Table 3: COPOD results for four different cases using various statistics.

| | OOD Dataset | | | |
|---|---|---|---|---|
| Statistic | SVHN | CIFAR100 | Hflip | Vflip |
| $u(\mathbf{x})$ | 0.96 | 0.59 | 0.54 | 0.59 |
| $v(\mathbf{x})$ | 0.94 | 0.56 | 0.54 | 0.59 |
| $w(\mathbf{x})$ | 0.93 | 0.58 | 0.54 | 0.61 |
| $v(\mathbf{x})$ & $w(\mathbf{x})$ | 0.94 | 0.58 | 0.54 | 0.60 |
| $u(\mathbf{x})$ & $v(\mathbf{x})$ | 0.97 | 0.61 | 0.53 | 0.61 |
| $u(\mathbf{x})$ & $w(\mathbf{x})$ | 0.98 | 0.61 | 0.54 | 0.61 |

Table 4: COPOD on individual statistics. IID dataset is CIFAR10.

In this dataset, $u(\mathbf{x})$ alone outperforms $v(\mathbf{x})$ in Table 3. We can see that ELBO's performance is somewhere between $u(\mathbf{x})$ and $v(\mathbf{x})$. This showcases the arithmetic cancellation discussed in Section 2. Our LPath method, in contrast, does not suffer from it and can combine their strengths to achieve stable and superior performances.

## D.2    INDIVIDUAL STATISTICS

To empirically validate how $(u(\mathbf{x}), v(\mathbf{x}), w(\mathbf{x}))$ complement each other suggested by Theorem 3.8, we use individual component alone in first stage and fit the second stage COPOD as usual. We notice signigicant drops in performances. We fit COPOD on individual statistics $u(\mathbf{x})$, $v(\mathbf{x})$, $w(\mathbf{x})$ and show the results in Table 4. We can see that our original combination in Table 1 is better overall.

## D.3    MD

To test the efficacy of $(u(\mathbf{x}), v(\mathbf{x}), w(\mathbf{x}))$ without COPOD, we replace COPOD by a popular algorithm in OOD detection, the MD algorithm Lee et al. (2018) and report such scores in Table 1. The scores are comparable to COPOD, suggesting $(u(\mathbf{x}), v(\mathbf{x}), w(\mathbf{x}))$ is the primary contributor to our performances.

## D.4    LATENT DIMENSIONS

One hypothesis on the relationship between latent code dimension and OOD detection performance is that lowering dimension incentivizes high level semantics learning, and higher level feature learning can help discriminate OOD v.s. IID. We conducted experiments on the below latent dimensions and report their AUC based on $v(\mathbf{x})$ (norm of the latent code) in Table 5

| Latent dimension | 1 | 2 | 5 | 10 | 100 | 1000 | 3096 | 5000 |
|---|---|---|---|---|---|---|---|---|
| $v(\mathbf{x})$ AUC | 0.39 | 0.63 | 0.52 | 0.45 | 0.22 | 0.65 | 0.76 | 0.59 |

Table 5: Lower latent code dimension doesn't help to discriminate in practice.

Clearly, lowering the dimension isn't sufficient to increase OOD performances.

