# OpenReview forum: "Rethinking Test-time Likelihood: The Likelihood Path Principle and Its Application to OOD Detection"
_ICLR.cc/2024/Conference — Submitted to ICLR 2024_

### Official Review · Reviewer_Jhho · 2023-10-27

**Soundness:** 2 fair
**Presentation:** 2 fair
**Contribution:** 3 good
**Rating:** 6
**Confidence:** 2

**Summary:**

This paper considers the OOD detection problem, where likelihood and scores either performs poorly or lack of provable guarantees. Under the VAE setup, the authors introduce the likelihood path (LPath) principle, suggesting that minimal sufficient statistics of VAEs’ conditional likelihoods are enough for OOD detection. Under several assumptions, the authors prove OOD detection guarantees for the chosen minimal sufficient statistics. Empirical results are also provided, suggesting the applicability of the proposed LPath principle.

**Strengths:**

The authors give a provable unsupervised OOD detection method that achieves good empirical performance, showing their work's high originality and significance. They also introduce several new concepts to facilitate the theoretical analysis.

**Weaknesses:**

1. I have the concern of whether the assumptions (essential separation concepts) are too strong, so that they can easily imply the theoretical guarantee. Also, I am not sure whether these "separations" are reasonable in realistic dataset.

2. The writing is unclear in the sense that some notation seems not to be defined, such as $p\_{\theta}$, $\mu\_z$. This makes me sometimes a little bit confusing.

**Questions:**

I am wondering whether the Definitions can be interpreted in a more standard way using conventional languages such as total variation distance (or some other distances) between OOD and IID distributions is large?

---

> ### Author Response · Authors · 2023-11-23
> **Response to Reviewer Jhho**
>
> > I have the concern of whether the assumptions (essential separation concepts) are too strong, so that they can easily imply the theoretical guarantee. Also, I am not sure whether these "separations" are reasonable in realistic dataset.
>
> We thank you for pointing this confusion out. We would like to invite you to Main Responses 7 and 8, and further clarify below.
>
> **[Essential separation is no stronger than prior literature]**
>
> The assumptions of essential separation and distance are no stronger than any other prior works, theoretical or empirical. To our best knowledge, Essential separation and distance strictly include “near” OOD and “far” OOD settings considered in most if not all prior literature [1, 2, 3]. If they were too strong, prior informal and rigorous attempts would all become too strong.
>
> We remark that prior arts consider either “far OOD” (total separation) or “near OOD” (not much separation). Essential separation is separation with a specified level of probability, which incorporates both. We invite you to explore the generality of our definitions via examples from B.2 to B.5.
>
>
> **[Our definitions work for any any metric including realistic datasets, e.g. SVHN v.s. CIFAR10]**
>
> They are also developed for any abstract metric spaces, in theory including human-like perceptual distances (proxied by perceptual distance in [4]). Take SVHN vs. CIFAR10 as an example, they are clearly separable (no one confuses digits with cars), as a result they are by definition essentially separable.
>
>
> **[Our proof mainly uses VAEs’ structure, not the essential separation]**
>
> As for whether these general assumptions can imply theoretical guarantees, we invite you to check our proof in the Appendix. First, our proof mostly uses VAEs’ unique structure, not the essential separation themselves. Second, since our definitions strictly include prior ones, prior assumptions are stronger. If the proof for VAEs simply follows from the definitions, prior works would have proved similar results. To our best knowledge, ours is the first provable work in the unsupervised OOD detection setting.
>
>
> >The writing is unclear in the sense that some notation seems not to be defined, such as $p_{\theta},  \mu_z$
> This makes me sometimes a little bit confusing.
>
> We thank you for improving our exposition. We assume familiarity to the VAEs literature where we recall Equation 1. $p_{\theta}$ is defined there. We also assumed familiarity with the minimal sufficient statistics of VAEs on page 4, where we defined the sufficient statistics $T$ including $\mu_z$. We will make these dependencies more explicit.
>
>
> [1] Zhen Fang, Yixuan Li, Jie Lu, Jiahua Dong, Bo Han, and Feng Liu. Is out-of-distribution detection learnable? Advances in Neural Information Processing Systems, 35:37199–37213, 2022.
>
> [2] Hendrycks, Dan, and Kevin Gimpel. "A baseline for detecting misclassified and out-of-distribution examples in neural networks." arXiv preprint arXiv:1610.02136 (2016).
>
> [3] Fort, Stanislav, Jie Ren, and Balaji Lakshminarayanan. "Exploring the limits of out-of-distribution detection." Advances in Neural Information Processing Systems 34 (2021): 7068-7081.
>
> [4] Gatys, Leon A., Alexander S. Ecker, and Matthias Bethge. "Image style transfer using convolutional neural networks." Proceedings of the IEEE conference on computer vision and pattern recognition. 2016.

---

### Official Review · Reviewer_hPfm · 2023-10-31

**Soundness:** 2 fair
**Presentation:** 1 poor
**Contribution:** 2 fair
**Rating:** 3
**Confidence:** 4

**Summary:**

The paper introduces a novel approach to out-of-distribution (OOD) detection in Variational Autoencoders (VAEs) by leveraging minimal sufficient statistics within the encoder and decoder. This method differs from Morningstar et al. (2022) by focusing on the mean and variance of $q(z|x)$ instead of the posterior entropy and KL divergence for the latent variable $z$. Theoretical guarantee is derived by assuming there is essential separation between in distribution and OOD samples w.r.t. the $L_2$ norm, and the encoder/decoder in VAEs are Lipschitz. But in practice the assumptions are not always realistic. The experiment results are sometimes surpass SOTA and sometimes perform worse when the assumptions broke.
1. **Proposed Methodology**:
  - The paper suggests the utilization of the minimal sufficient statistics in both the encoder and decoder of the VAE for OOD detection.
  - The work is reminiscent of Morningstar et al. (2022), but distinguishes itself by emphasizing the mean and variance of $q(z∣x)$ and derive theoretical guarantee of OOD under assumptions.
2. **Assumptions and Implications**:
  - Essential separation of in-distribution (ID) and OOD data based on L2 norm distance is assumed.
  - The encoder and decoder must satisfy Lipschitz type conditions.
  - Under these conditions, detection using reconstruction error or L2 norm distance in the latent space between a sample and ID samples is reliable with high probability.
    - Practical implementation does not calculate distance by sampling IID samples. Instead, it is approximated using $\mid || \mu_{{z}}\left({x}_{\mathrm{OOD}}\right) \|-r_0 \mid$.
3. **Experimental Outcomes**:
  - The method proves effective when the assumption on essential separation is likely met.
  - On datasets with minor separations, like horizontally and vertically flipped variants, the technique is less effective compared to some state-of-the-art (SOTA) methods.

**Strengths:**

- The proposed sufficient statistics used for OOD detection are different from Morningstar et al. (2022) by focusing on the mean and variance of $q(z|x)$ instead of the posterior entropy and KL divergence for the latent variable $z$.

**Weaknesses:**

- The utilization of  $\mid || \mu_{z} (x_{{OOD}}) \|-r_0 \mid$ to approximate the distance between test and ID samples is questioned for its lack of a principled basis. If ID samples have a wide spread in $μ(x_{IID})$ in the latent space, the absence of a singular reference point makes the approximation meaningless. The reliance on VAEs' regularization of the posterior on $z$ towards a Gaussian (typically zero mean) implies the technique may not be generalizable to other generative models with distinct latent variable regularization.

- The assumptions for essential separation and Lipschitz conditions are too strong. The separation (defined by the $L_2$ norm)  may not hold for real world problems. The Lipschitz conditions are not enforced during training of VAEs (or in other generative models) as well.

- The idea of using reconstruction error for OOD detection was proposed in [1]. It is worth discussing the difference and what are the new interpretations.

- The paper's presentation quality needs improvements.

  - Some concepts are articulated in an imprecise, non-rigorous manner. For example, the phrase “break in the right way” from Section 2 lacks clarity.
  - Several explanations are relegated to appendices, compromising the fluidity and comprehension of the main text. Definitions, like B.6 and B.7, are cited without main text elaboration.
  - The excessive use of bold text and protracted informal subtitles detract the reader.

[1] Osada, Genki, Tsubasa Takahashi, Budrul Ahsan, and Takashi Nishide. "Out-of-Distribution Detection with Reconstruction Error and Typicality-based Penalty." In Proceedings of the IEEE/CVF Winter Conference on Applications of Computer Vision, pp. 5551-5563. 2023.

**Questions:**

- When the assumptions will hold and measure it empirically if possible to validate?
- Explain the use of $\mid || \mu_{z} (x_{{OOD}}) \|-r_0 \mid$, when does this serve as a good approximate? Is this limited to methods like VAE that regularize posterior distribution of $z$?

---

> ### Author Response · Authors · 2023-11-23
> **Response to Reviewer hPfm**
>
> >The utilization of  |$\lvert \mu_{z}(x_{\text{OOD}}) - r_0 \rvert$| to approximate the distance between test and ID samples is questioned for its lack of a principled basis. If ID samples have a wide spread in in the latent space, the absence of a singular reference point makes the approximation meaningless.
>
> Thank you for pointing this our and improving our rigor. We invite you to Main Response 8 to see our now fully rigorous justification. In particular, the aforementioned approximations, as in Equations 13 and 14, enjoy similar provable guarantees (Proposition B 16 and Corollary B 17) as Theorem 3.8. The gap between Equation 10 (main theorem) and Equations 13 +14 (experimental version) depends on empirically verifiable concentration phenomenon (discussed in Section 3.2, illustrated on Figure 2. In light of these observations, we also strengthen this concentration via regularization (Section 3.2)).
>
> For the DC-VAEs we used, even without regularization, we empirically observe that the latent code concentrates. You are correct that when IID samples have a wide spread, our experiments without regularization showed minor performance degradation. But the unregularized model still delivers strong performances.
>
> Theoretically, this is related to our essential definitions. Many false statements in fact are true with high probability. The widespread latent code is an example: while the latent code norm can spread a wide range, most of them are concentrated near a narrow range. In sum, widespread latent codes might weaken an absolute logical statement to a high probability one. But it does not make it meaningless. These are experimentally verified and rigorously proved (Proposition B 16 and Corollary B 17).
>
>
> >The reliance on VAEs' regularization of the posterior on towards a Gaussian (typically zero mean) implies the technique may not be generalizable to other generative models with distinct latent variable regularization.
>
> You are correct that the regularization is VAEs specific, but we would like to invite you to visit Main Response 4 for why our VAEs model specific techniques are useful in the present unsupervised OOD detection we consider.
> The no free lunch principle, stating no single algorithm works the best in all cases, also discussed in Footnote 3, suggests dependence on VAEs can also be a strength, making our dependence on VAEs a double-edged sword. Whether it is a weakness or not highly depends on the applications of interest.
>
> When a few current SOTA sits below AUC 0.7, we are willing to trade off generality for empirical performances. Looking at Table 1, we believe taking advantage of VAEs’ structural uniqueness is exactly why our empirical performances match or exceed SOTA benchmarks which are based on arguably more sophisticated models (Glow, DDPM, etc.). This implicitly shows our method is more sophisticated, since our model is much smaller. The sophistication probably comes from our model specific choice. For the present application (IID v.s. OOD), while the benefits of richer latent structures remain unclear (geometrically, we only need to carve the latent or visible spaces into IID v.s. OOD, it is less clear richer latent structure surely helps), our methods’ performances are demonstrated.
>
> Moreover, as discussed previously, our performances only degraded a little without such concentration regularizations. This is because the unregularized VAEs already exhibit concentration phenomenon when VAEs have high latent dimensions, which motivates us to further strengthen it.

---

> > ### Author Response · Authors · 2023-11-23
> > **Response II**
> >
> > > The assumptions for essential separation and Lipschitz conditions are too strong. The separation (defined by the L2 norm) may not hold for real world problems.
> >
> > We thank you for pointing out these expository improvements.
> >
> > For a discussion on essential separation, we would like to invite you to visit Main Response 7. In short, The assumptions of essential separation and distance are no stronger than any other prior works, theoretical or empirical. To our best knowledge, Essential separation and distance strictly include “near” OOD and “far” OOD considered in most if not all prior literature [1, 2, 3]. If they were too strong, prior informal and rigorous attempts would all become too strong.
> >
> >
> > In our construction of essential separation, the probabilistic guarantee $\epsilon_{\text{IID}}$ and $\epsilon_\text{OOD}$ trades off with the margin $m_{\text{inter}}$. For a difficult problem (“near OOD” such as HFlip), if we want a big margin, the probabilities won’t be high. We invite you to Appendix B.1, Examples B.4 and B.5 for illustrations.
> > In there, we show how essential separation and distance behave even when $P_{\text{IID}} = P_{\text{OOD}}$. In short, it would produce claims like, with probability at least greater than 0, $P_{\text{IID}} = P_{\text{OOD}}$ are separated by a fixed positive margin. However, when $P_{\text{IID}}$ and $ P_{\text{OOD}}$ begin to isolate a little, it would say, with probability greater than 0.1, $P_{\text{IID}} = P_{\text{OOD}}$ are separated by 0.01. The harder the problem, the more vacuous the guarantees become. The more separable between $P_{\text{IID}}$ and $ P_{\text{OOD}}$, the bigger their essential distance (far OOD problem such as SVHN v.s. CIFAR).
> >
> >
> > For the L2 norm, we invite you to Main Response 8. In short, our proof techniques do not rely on the L2 norm property; it holds for any metric spaces (e.g. $L^{\infty}$ popular in the adversarial robustness literature, perceptual distance [4], etc.).
> >
> > We chose L2 norm for the paper because of VAEs' Gaussian decoder. We agree that the utility of Theorem 3.8 depends on which perceptual distance is used. However, designing a proper metric for perception is beyond the scope of the paper - we only prove that our theory likely includes it. We do not consider this as a weakness of our paper. We will restate and modify the proof of Theorem 3.8 for any abstract metric spaces.
> >
> >
> > > The Lipschitz conditions are not enforced during training of VAEs (or in other generative models) as well.
> >
> > We thank you for this insightful comment. We have done some works on it, but we did not include it in the paper, due to the ~20 pages appendix.
> >
> > Lipschitz conditions can be enforced ([5], [6], [7], [8]]) during VAEs training, and we conducted preliminary experiments on enforcing the decoder. However we do not observe substantial differences on OOD detection performances. The problem with this constraint alone is that Theorem 3.8 also requires the encoder to be co-Lipschtiz (our new definition) as well. Since this involves the inverse image of the encoders, we are unaware of any literature that enforces co-Lipscchtizness.
> >
> > We managed to link it to some better known concepts, i.e. quasi-isometry in geometric group theory (Co-Lipschitzness and Quasi-Isometric Embedding, Section B.2 in Appendix). We reformulated co-Lipschitzness in a way that is reminiscent of the adversarial robustness certification literature. We remark that it took the adversarial robustness community a few years before this became more mature, in the supervised classification setting. Enforcing both Lipschitz and co-Lipscthiz is highly non-trivial in our setting and is beyond the scope of the present paper.
> >
> > Nevertheless, even if we are unable to enforce it, it does not mean DGMs are not Lipschitz or co-Lipschitz. In fact, empirical evidence suggests that they are ([9]), but the constants can be larger than ideal. We will add these observations to the Appendix.
> >
> >
> > > The paper's presentation quality needs improvements. Some concepts are articulated in an imprecise, non-rigorous manner. For example, the phrase “break in the right way” from Section 2 lacks clarity.
> >
> > We intended to make the paper more lively due to relatively more math in the paper. “Break in the right way” is detailed in Section B.4.2. We basically mean all the known VAEs likelihood estimation is broken, but how we pair two broken VAEs can lead to performance gain.
> >
> > > Several explanations are relegated to appendices, compromising the fluidity and comprehension of the main text. Definitions, like B.6 and B.7, are cited without main text elaboration. The excessive use of bold text and protracted informal subtitles detract the reader.
> >
> > While our paper develops some new theories, we also want it to be accessible to ML practitioners. This will create some exposition difficulty. We thank your suggestion and will try to improve the flow.

---

> > > ### Author Response · Authors · 2023-11-23
> > > **Response III**
> > >
> > > > When the assumptions will hold and measure it empirically if possible to validate?
> > >
> > > We thank you for this insightful comment. We have now included a new Section, Co-Lipschitzness and Quasi-Isometric Embedding, in the Appendix B.2. As discussed in our previous comment on enforcing Lipschtizness, we relate co-Lipschtizness and quasi-isometry in geometric group theory. This suggests ways of estimating the co-Lipschitz degrees, for example, by rethinking the adversarial robustness certification literature (both lower and upper bounds). We believe this is very interesting, but it is beyond the scope of the present paper.
> > >
> > > To our best knowledge, this is the first provable work in the DGMs unsupervised OOD setting. While imperfect, we believe it adds significant value to the current works. Fully verifying it requires much more not-yet-developed machinery, which could require a few papers of work.
> > >
> > > > Explain the use of |$\lvert \mu_{z}(x_{\text{OOD}}) - r_0 \rvert$|  when does this serve as a good approximate? Is this limited to methods like VAE that regularize posterior distribution of ?
> > >
> > > We believe this is already addressed in the above discussion (Proposition B 16 and Corollary B 17) where the approximation gap depends on the empirically verifiable concentration effect, but we reiterate that the no free lunch principle applies. A method that is “limited” to the standard VAEs also means this method is tailored for it. This inductive bias makes the method perform well in our applications of interest, namely OOD detection. If any OOD detection score function is applicable to all other models, it is unlikely to perform well in many detection problems. See footnote 3 in our paper or [10] for more details.
> > >
> > > [1] Zhen Fang, Yixuan Li, Jie Lu, Jiahua Dong, Bo Han, and Feng Liu. Is out-of-distribution detection learnable? Advances in Neural Information Processing Systems, 35:37199–37213, 2022.
> > >
> > > [2] Hendrycks, Dan, and Kevin Gimpel. "A baseline for detecting misclassified and out-of-distribution examples in neural networks." arXiv preprint arXiv:1610.02136 (2016).
> > >
> > > [3] Fort, Stanislav, Jie Ren, and Balaji Lakshminarayanan. "Exploring the limits of out-of-distribution detection." Advances in Neural Information Processing Systems 34 (2021): 7068-7081.
> > >
> > > [4] Gatys, Leon A., Alexander S. Ecker, and Matthias Bethge. "Image style transfer using convolutional neural networks." Proceedings of the IEEE conference on computer vision and pattern recognition. 2016.
> > >
> > > [5] Barrett, Ben, et al. "Certifiably robust variational autoencoders." International Conference on Artificial Intelligence and Statistics. PMLR, 2022.
> > >
> > > [6] Zhang, Bohang, et al. "Boosting the Certified Robustness of L-infinity Distance Nets." International Conference on Learning Representations. 2021.
> > >
> > > [7] Anil, Cem, James Lucas, and Roger Grosse. "Sorting out Lipschitz function approximation." International Conference on Machine Learning. PMLR, 2019.
> > >
> > > [8] Zhang, Huan, Pengchuan Zhang, and Cho-Jui Hsieh. "Recurjac: An efficient recursive algorithm for bounding the Jacobian matrix of neural networks and its applications." Proceedings of the AAAI Conference on Artificial Intelligence. Vol. 33. No. 01. 2019.
> > >
> > > [9] Fazlyab, Mahyar, et al. "Efficient and accurate estimation of lipschitz constants for deep neural networks." Advances in Neural Information Processing Systems 32 (2019).
> > >
> > > [10] Noah Simon and Robert Tibshirani. Comment on" detecting novel associations in large data sets" by reshef et al, science dec 16, 2011.

---

### Official Review · Reviewer_tps4 · 2023-11-04

**Soundness:** 2 fair
**Presentation:** 1 poor
**Contribution:** 2 fair
**Rating:** 5
**Confidence:** 3

**Summary:**

The submission #8717  presents a new perspective on out-of-distribution detection with the introduction of the "Likelihood Path Principle". The principle is based on the observation that traditional likelihood measures can be ineffective for OOD detection due to their reliance on static data snapshots. The authors suggest a dynamic path-wise likelihood integration method to capture the evolving nature of data distributions.

The paper asserts that this method more accurately differentiates between in-distribution (ID) and OOD samples by considering the trajectory of the likelihood as data moves from ID to OOD. To substantiate their claims, the authors provide experimental results that demonstrate an improvement over existing methods such as ODIN across several benchmarks. Additionally, they offer theoretical insights into why considering the path of likelihood can be beneficial for OOD detection.

**Strengths:**

- The paper introduces new theoretical tools (section 2, section3) that provide a solid foundation for their proposed LPath principle and OOD detection approach. Unlike some previous work, the proposed method comes with non-asymptotic provable guarantees for OOD detection.

- The paper claims state-of-the-art empirical results, suggesting a significant advancement over existing methods.

**Weaknesses:**

- *Complexity of implementation*. While not explicitly mentioned, the introduction of new theoretical concepts might imply a more complex implementation and understanding, potentially limiting accessibility for practitioners. The author has not provided any valid implementations for reviewing.

- *Dependence on VAEs*. The method's effectiveness may be highly dependent on the performance and tuning of the underlying VAEs, which can be sensitive to hyperparameters and data quality.

- *Presentation*. I feel that the presentation quality of the manuscript could still be improved, especially some illustrations/figures are difficult to read.

Additionally, please refer to the ‘Questions’ section for my other potential concerns.

**Questions:**

- What is the computational overhead introduced by the path-wise likelihood calculation, and how does it scale with the complexity of the model and the size of the dataset?

- Continuing above, in high-dimensional spaces, traditional likelihood methods often struggle due to the curse of dimensionality. How does the Likelihood Path Principle mitigate these issues, and is there a threshold where the method becomes computationally infeasible?

- How does the LPath algorithm perform under different types of data distributions and noise levels? How does the method handle cases where the OOD data is deliberately designed to mimic ID data, as in adversarial attacks?

- Can the principles introduced in the paper be extended to other types of generative models beyond VAEs? Also, is there potential for the Likelihood Path Principle to be integrated into a wider array of model architectures beyond those tested, including unsupervised and semi-supervised learning scenarios?

---

> ### Author Response · Authors · 2023-11-23
> **Response to Reviewer tps4**
>
> > Complexity of implementation. While not explicitly mentioned, the introduction of new theoretical concepts might imply a more complex implementation and understanding, potentially limiting accessibility for practitioners. The author has not provided any valid implementations for reviewing.
>
> We did not provide implementations because there is dependency on private repos. We have attached a self-contained jupyter notebook that illustrates the pipeline on synthetic datasets. We would like to emphasize that although it takes some math to derive our algorithm, its implementation is simple, similar to DoSE [1] as we mentioned in the paper.
>
>
> > Dependence on VAEs. The method's effectiveness may be highly dependent on the performance and tuning of the underlying VAEs, which can be sensitive to hyperparameters and data quality.
>
> We would like to invite you to visit Main Response 4 for our choice of VAEs based algorithm, per no free lunch principle. We believe model specific OOD detection is advantageous when a few SOTA sits below AUC 0.7. The no free lunch principle is also discussed in Footnote 3, which suggests dependence on VAEs can also be a strength, making our dependence on VAEs a double-edged sword. Whether it is a weakness or not highly depends on the applications of interest.
>
> On one hand, you are right that this dependence makes our detection algorithm less general. On the other hand, we believe taking advantage of VAEs’ structural uniqueness is exactly why our empirical performances match or exceed SOTA benchmarks which are based on arguably more sophisticated models (Glow, DDPM, etc.). This implicitly shows our method is more sophisticated, since our model is much smaller.
>
> We note that however that the proposed LPath principle is general and orthogonal to improving VAEs. This showcases the opportunity to conduct research and improve upon the aforementioned directions independently.
>
> Last but not the least, under practical constraints, it can be difficult to further improve the quality of many generative models (e.g. their robustness to hyperparameters, etc. See Introduction, the works of Behrmann et al., 2021; Dai et al.) in practice. When we cannot further improve the underlying model, our method can still yield significant improvement on the OOD detection performance (we performed targetted experiments to empirically verify it in Table 3, Section D.1, in Appendix). We believe this is our method’s added value and it is orthogonal to model improvements, and it is one of the main motivations behind this work.
>
>
> [1] Morningstar, Warren, et al. "Density of states estimation for out of distribution detection." International Conference on Artificial Intelligence and Statistics. PMLR, 2021.

---

> > ### Author Response · Authors · 2023-11-23
> > **Response to Reviewer tps4 - II**
> >
> > > What is the computational overhead introduced by the path-wise likelihood calculation, and how does it scale with the complexity of the model and the size of the dataset?
> >
> > There is no non-standard computational overhead.
> >
> > In the first neural feature extraction stage, the method requires no more calculation than forward propagation in all other DGMs. Our method doesn’t affect regular neural net training, and so it only depends on how big the VAEs (or other DGMs) are.
> >
> > In the second classical statistical stage, it depends on the second stage algorithm chosen, as you correctly pointed out. For the more recent methods, such as COPOD, it scales linearly with respect to the number of extracted features in the first stage (=3 in the present paper), and this second stage can handle datasets with 10,000 features and 1,000,000 observations on a standard personal laptop.
> >
> > > Continuing above, in high-dimensional spaces, traditional likelihood methods often struggle due to the curse of dimensionality. How does the Likelihood Path Principle mitigate these issues, and is there a threshold where the method becomes computationally infeasible?
> >
> > Computationally, if we use COPOD, LPath scales very well, as stated above. Methodologically, since our LPath principle is a hybrid based on neural models, the first neural feature extraction stage helps with the curse of dimensionality. In our case, thanks to Theorem 3.8, we further distill only 3 features from the minimal sufficient statistics. Therefore, the second stage traditional method remains uncursed for such low dimensions.
> >
> > > How does the LPath algorithm perform under different types of data distributions and noise levels? How does the method handle cases where the OOD data is deliberately designed to mimic ID data, as in adversarial attacks?
> >
> > We thank you for bringing up this interesting point.
> >
> > Theoretically speaking, our method does not utilize any dataset information, unlike some other works on image datasets, and thus can be extended to other types of data. However this is beyond the scope of this paper, as we are only taking a first step at establishing the theoretical and empirical foundation for the LPath method. We invite you to Main Response 3 for our motivation.
> >
> > Considering adversarial attacks as OOD is extremely interesting, but it is beyond the scope of our paper and all SOTA works we compared to. The present paper tackles the “natural” OOD setting.
> >
> >
> > > Can the principles introduced in the paper be extended to other types of generative models beyond VAEs? Also, is there potential for the Likelihood Path Principle to be integrated into a wider array of model architectures beyond those tested, including unsupervised and semi-supervised learning scenarios?
> >
> > We thank you again for raising this point. We have mentioned extending the LPath principle to other DGMs. We are indeed working on such generalizations in a follow up work, on Glow and DDPM. However, these more advanced models possess fewer statistical structures, making the theoretical analysis trickier. We are hopeful that such an extension will lead to even better performances on the OOD detection benchmarks.
> >
> > We believe there is a potential for the LPath principle to be extended, as long as likelihood plays an important role in the problem setting or task, which is fairly common. We invite you to visit Main Response 5 and 6 for potential extensions to semi-supervised and supervised settings, as well as applications to other DGMs.

---

### Official Review · Reviewer_6idB · 2023-11-06

**Soundness:** 3 good
**Presentation:** 3 good
**Contribution:** 2 fair
**Rating:** 6
**Confidence:** 2

**Summary:**

The paper develops a generalization of Likelihood principle that comes with provable guarantees for OOD detection in deep generative models. Applying this new principle to VAEs, the authors propose using minimal sufficient statistics for OOD detection with non asymptotic guarantees. Empirical results show the suggested approach can outperform or perform on par with other OOD detection methods in an unsupervised setting.

**Strengths:**

- The paper analyzes different types of OOD samples and the reason behind the difficulty of some of OOD cases in a principled way.
- It provides a theory and a simple computational approach for identifying OODs.

**Weaknesses:**

- The intuition behind the theorem and the illustration in Figure 1 can be further improved. As the main figure of the paper which is introducing the idea, Figure 1 is not easy to follow. You need to read the paper all the way to the end of page 7 so you can understand the 4 cases and their connection to the idea presented in the paper.
- As the authors have mentioned, Equation 10 is non-trivial to compute and an approximation is provided. The effect of the error of this approximation on the performance of the algorithm can be further studied in synthetic cases.
- Setting the decision criteria in the proposed algorithm is non-trivial and can be further studies in the paper.
- The fact that the method outperforms other VAE baselines but doesn’t perform as well as more sophisticated baselines makes the practical usage of the method in safety critical domains less feasible.

**Questions:**

- Minor: Fix the references to Equations 19 and 18 in section 3.2.
- Fix “∥x_OOD − x_OOD∥_2 is large” on page 7
- How does the approximation error of Equation 12 affect the performance?
- The motivation behind the paired VAE idea in section 4 is unclear. The idea has been introduced in few lines and the reasoning behind it is deferred to the appendix. Can you either expand the motivation in the main text or move these few lines to the appendix?
- What are the hyperparameters that are needed for the OOD decision rule?
- Minor: I think the citations for DDPM and LMD are flipped

---

> ### Author Response · Authors · 2023-11-23
> **Response to Reviewer 6idB - I**
>
> > The intuition behind the theorem and the illustration in Figure 1 can be further improved. As the main figure of the paper which is introducing the idea, Figure 1 is not easy to follow. You need to read the paper all the way to the end of page 7 so you can understand the 4 cases and their connection to the idea presented in the paper.
>
> We thank you for pointing this out. We will work on a revised figure with modified caption to better illustrate the LPath principle.
>
> > As the authors have mentioned, Equation 10 is non-trivial to compute and an approximation is provided. The effect of the error of this approximation on the performance of the algorithm can be further studied in synthetic cases.
>
> We believe you are referring to the gap between Equation 10 and Equations 13+14. We have now provided a more detailed justification with rigorous proofs. Equation 13 and 14 now come with similar guarantees (Proposition B 16 and Corollary B 17). The gap between Equation 10 and Equations 13+14 depends on empirically verifiable concentration phenomenon (discussed in Section 3.2, illustrated on Figure 2). In light of these observations, we also strengthen this concentration via regularization (Section 3.2).
>
>
> > Setting the decision criteria in the proposed algorithm is non-trivial and can be further studied in the paper.
>
> We are unsure what criteria Reviewer 6idB refers to. We believe this is related to the hyperparameters needed for the OOD decision rule, in the Question section. If we are wrong, we would like to ask for more elaborations. Note that we also discussed training and inference hyperparameters/configurations in Appendix C.3. Our neural feature extraction stage (Section 4) is quite standard, except that we tune latent dimensions (Appendix B 4.2, C.3). Our second stage statistical algorithm requires no hyperparameter.
>
> > The fact that the method outperforms other VAE baselines but doesn’t perform as well as more sophisticated baselines makes the practical usage of the method in safety critical domains less feasible.
>
> We would like to emphasize that while we do not perform better in all cases, we surpass or match all others except SVHN (IID) vs CIFAR 10/VFlip/HFlip (OOD). Among these mostly widely used benchmarks, our method is at least as good as all others overall.
>
> As for practical application in safety critical domains, we invite you to Main Response 2 for our advantage. In safety critical domains, theoretical guarantees are arguably more important in OOD detection, because in deployment, there is no way to control the streaming data. Offline validations such as the ones every paper performs are not guarantees to translate to the real world.
>
> In contrast, as a provable method, even if we are not the best in the deployment environment, we can theoretically identify the factors that caused the degradation. The co-Lipschitz degrees (K, k) are the most difficult to estimate in practice, which is a separate topic itself. We now developed Lemma B.12 that suggests a way of estimating those, related to the adversarial robustness certification literature.
>
> For the above reasons, we believe our method is more interpretable and reliable than other SOTA methods. We would like to hear from reviewer 6idB whether this discussion has addressed the aforementioned concern.
>
> > How does the approximation error of Equation 12 affect the performance?
>
> Approximation occurs in Equation 13 and 14, where justification heuristically is provided in Footnote 10 and fully rigorously in Proposition B 16 and Corollary B 17 in the new version. They also come with provable guarantees.
>
> > The motivation behind the paired VAE idea in section 4 is unclear. The idea has been introduced in few lines and the reasoning behind it is deferred to the appendix. Can you either expand the motivation in the main text or move these few lines to the appendix?
>
> We thank you for improving our exposition. We propose adding the following to Section 4. The idea behind VAEs pairing is that Equation 10 in Theorem 3.8 prefers smaller K, while Equation 11 prefers bigger K. These two conditions cannot be satisfied by a single VAE, but can be achieved with a paired VAEs. More details and discussion are given in the Appendix B 4.2 and B 4.3.
>
> We appreciate your additional thoughts on the change. Any further suggestions would be welcomed.

---

> > ### Author Response · Authors · 2023-11-23
> > **Response to Reviewer 6idB - II**
> >
> > > What are the hyperparameters that are needed for the OOD decision rule?
> >
> > We do not require any hyperparameter in our paper as we used the AUROC as the evaluation metric, commonly used in the literature. We may be missing something basic, could you clarify what decision criteria are being referred to?
> >
> > Our algorithm (Page 8) and Appendix C.3 VAE ARCHITECTURE AND TRAINING describe how we get a decision score for OOD detection.
> >
> > In stage 1, our VAEs training follows the widely cited open sourced implementation of DC-VAEs, and we train in a similar manner. The major difference is that we experiment with different latent dimensions guided by our theory.
> >
> > In stage 2, from the open sourced implementation, https://github.com/winstonll/COPOD/blob/master/models/cod.py, since COPOD requires no hyper-parameter to fit the 2nd stage classical OOD algorithm, it is easy to obtain a decision score.

---

### Author Response · Authors · 2023-11-23
**Main Responses**

We thank all the reviewers for the insightful feedback. We address the common confusion and concerns here, and individually below.

**Main Responses**

**1. [Real world OOD cannot be simulated makes our provable method advantageous]**

Provable performances are arguably more important than IID setting, as *real world OOD performances cannot be experimentally validated because we do not have access to the OOD data in advance*. This makes the provable methods advantageous against the less principled ones in both performance and interpretability, in the safety critical settings. To our best knowledge, this is the first provable method to bridge this gap. While imperfect, it is a good first step.

**2. [Advancing a SOTA is a clear contribution]**

In four of the most widely used benchmarks, we have achieved significant progress. We established a new State of the Art (SOTA) in one, matched two others, and are close to the top in the remaining benchmark. This achievement of a new SOTA represents a clear empirical contribution.

**3. [We introduce LPath by showcasing one setting well]**

We share reviewers’ eagerness to extend LPath to other settings. We think that this principle of considering the sufficient statistics of likelihood as a trajectory is interesting enough on its own, and could potentially open up many future directions. This is why we want to make a solid introduction to the LPath principle with an in-depth theoretical analysis and thorough empirical evidence in the standard vision setting, for unsupervised OOD detection. While the LPath principle has the potential for other data types and models, we believe that when proposing a new principle, focusing on a concrete use case can bring greater benefit to the community, as it provides a solid ground for future works.

**4. [VAEs structure specific algorithm can be an advantage, per no free lunch trade off]**

The no free lunch principle reminds us that there is no one single algorithm that does well in all settings. While the Lpath principle is very general and applicable in many settings, when we instantiate it to our specific setting, we can further leverage the structure of VAEs for further performance gain, and this is important when current SOTA lacks. In our case, the unsupervised OOD detection problem still has a few SOTA sitting below 0.7 AUC, advancing empirical performances is quite meaningful. Our VAEs model specific design makes our OOD detection algorithm comparable to SOTA that use much bigger models (Glow, DDPM, etc.). We do not consider such specific choices to be a weakness in the present work but a trade off we are willing to make, per no free lunch. When encountering a different application, our algorithm will change accordingly.

**5. [LPath for (semi-)supervised learning]** The LPath principle is a versatile tool whenever the models are imperfect, beyond the unsupervised setting considered in the paper. In VAEs’ semi-supervised learning setting [1, 2], the LPath principle can be applied to study the relation between labeled + unlabeled LPaths and the confidence calibration/statistical generalization errors [3, 4]. In the supervised setting, LPath can track activation paths that lead to the SoftMax probabilities, although this requires more developments.

However, we believe studying in-depth on one concrete application, namely unsupervised OOD detection, is a better introduction to this new principle (resulting in more than 20 pages of derivations, analysis and discussion in Appendix). We leave additional applications of the LPath principle to future works.

**6. [LPath for more DGMs]**

The LPath principle can be adapted to the other two DGMs (Glow and DDPM) popular in the OOD detection literature. Theoretically, it applies to any DGMs that forward propagate to likelihoods. This will be showcased in future works currently undergoing.

**7. [Essential separation is no stronger than prior literature]**

Essential separation and distance strictly include “near” OOD and “far” OOD considered in most if not all prior works [5, 7, 8]. If it was too strong, prior informal and rigorous attempts would all become too strong. In our construction, the probabilistic guarantee $\epsilon_\text{IID}$ and $\epsilon_\text{OOD}$ trades off with the margin $m_{\text{inter}}$. For a difficult problem (“near OOD” such as HFlip), if we want a big margin, the probabilities won’t be high. For easier ones (“far OOD” such as SVHN v.s. CIFAR), we get high probability guarantees for a non-trivial margin $m_{\text{inter}}$.

---

> ### Author Response · Authors · 2023-11-23
> **Main Responses Continued**
>
> **8. [Our main theorem holds for any metric space; our approximation is now fully justified]**
>
> Theorem 3.8 in fact generalizes to all metric spaces ($L^{\infty}$ in adversarial robustness, perceptual distance in vision [6], etc.). We chose the L2 norm because of the Gaussian decoder and its accessibility. Our exposition aimed to limit the introduction of new math and hence the L2 choice.
> We believe our approximations of Equation 10 by Equations 13 and 14, are sufficiently intuitive, but thanks to the reviewing process, we have now made them fully rigorous with similar guarantees depending on an empirically verifiable gap. The new statements and proofs are added for all above (Theorem 3.8, Proposition B 16 and Corollary B 17).
>
> [1] Kingma, Durk P., et al. "Semi-supervised learning with deep generative models." Advances in neural information processing systems 27 (2014).
>
> [2] Feng, Hao-Zhe, et al. "Shot-vae: semi-supervised deep generative models with label-aware elbo approximations." Proceedings of the AAAI Conference on Artificial Intelligence. Vol. 35. No. 8. 2021.
>
> [3] Guo, Chuan, et al. "On calibration of modern neural networks." International conference on machine learning. PMLR, 2017.
>
> [4] Minderer, Matthias, et al. "Revisiting the calibration of modern neural networks." Advances in Neural Information Processing Systems 34 (2021): 15682-15694.
>
> [5] Zhen Fang, Yixuan Li, Jie Lu, Jiahua Dong, Bo Han, and Feng Liu. Is out-of-distribution detection learnable? Advances in Neural Information Processing Systems, 35:37199–37213, 2022.
>
> [6] Gatys, Leon A., Alexander S. Ecker, and Matthias Bethge. "Image style transfer using convolutional neural networks." Proceedings of the IEEE conference on computer vision and pattern recognition. 2016.
>
> [7] Hendrycks, Dan, and Kevin Gimpel. "A baseline for detecting misclassified and out-of-distribution examples in neural networks." arXiv preprint arXiv:1610.02136 (2016).
>
> [8] Fort, Stanislav, Jie Ren, and Balaji Lakshminarayanan. "Exploring the limits of out-of-distribution detection." Advances in Neural Information Processing Systems 34 (2021): 7068-7081.

---

### Meta-Review · Area_Chair_8rv1 · 2023-12-15

**Metareview:**

The paper presents a novel approach to out-of-distribution (OOD) detection in deep generative models, particularly focusing on Variational Autoencoders (VAEs) and empirical results in OOD detection using VAEs. However, the proposed method's main result, complexity, and practical applicability in certain domains need to be clarified. The paper may need improvement in presenting key concepts and further exploring the algorithm's implementation details and assumptions. The proof of the main theorem seems to be stylized and lacks rigor.

**Justification For Why Not Higher Score:**

The paper needs further improvement in various aspects.

**Justification For Why Not Lower Score:**

NA

---

### Decision · Program_Chairs · 2024-01-16

Reject